# Dual-Res Tandem Mamba-3D: Bilateral Breast Lesion Detection and Classification on Non-contrast Chest CT

**Jiaheng Zhou**♥♦♠§*    **Wei Fang**♦♣Ψ*    **Luyuan Xie**Θ♦    **Yanfeng Zhou**Ω
**Lianyan Xu**Π    **Minfeng Xu**♠♣    **Ge Yang**♥♦†    **Yuxing Tang**♦†

♥Institute of Automation, Chinese Academy of Sciences
♦School of Artificial Intelligence, University of Chinese Academy of Sciences
♠DAMO Academy, Alibaba Group    ♣Hupan Laboratory, Hangzhou, China
ΨZhejiang University    ΘPeking University
ΩShenzhen University    ΠZhongShan Hospital, Fudan University
§Work done during an internship at the Medical AI Lab, DAMO Academy, Alibaba Group.
*Equal contribution.    †Corresponding authors.
{zhoujiaheng2022, ge.yang}@ia.ac.cn    {lucas.fw, yuxing.t}@alibaba-inc.com

## Abstract

Breast cancer remains a leading cause of death among women, with early detection significantly improving prognosis. Non-contrast computed tomography (NCCT) scans of the chest, routinely acquired for thoracic assessments, often capture the breast region incidentally, presenting an underexplored opportunity for opportunistic breast lesion detection without additional imaging cost or radiation. However, the subtle appearance of lesions in NCCT and the difficulty of jointly modeling lesion detection and malignancy classification pose unique challenges. In this work, we propose **D**ual-**R**es **T**andem **M**amba-**3D** (**DRT-M3D**), a novel multitask framework for opportunistic breast cancer analysis on NCCT scans. DRT-M3D introduces a dual-resolution architecture, which captures fine-grained spatial details for segmentation-based lesion detection and global contextual features for breast-level cancer classification. It further incorporates a tandem input mechanism that models bilateral breast regions jointly through Mamba-3D blocks, enabling cross-breast feature interaction by leveraging subtle asymmetries between the two sides. Our approach achieves state-of-the-art performance in both tasks across multi-institutional NCCT datasets spanning four medical centers. Extensive experiments and ablation studies validate the effectiveness of each key component.

## 1 Introduction

Breast cancer remains one of the most prevalent and deadly diseases among women worldwide [8]. Early detection is critical for improving survival outcomes [74]. While screening techniques such as mammography, ultrasound, and magnetic resonance imaging (MRI) are well established, they require dedicated protocols and are not routinely performed during thoracic exams for unrelated conditions [13]. However, non-contrast chest computed tomography (chest NCCT) is widely available and frequently conducted for pulmonary evaluation, which naturally includes the breast region [61]. As illustrated in Fig. 1, this creates a valuable opportunity for breast cancer assessment without additional imaging cost or radiation exposure.

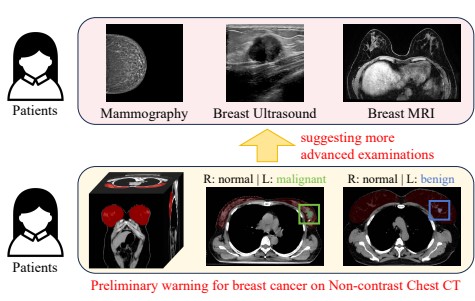

Figure 1: Non-contrast chest CTs enable opportunistic breast cancer analysis without additional cost. Leveraging bilateral information leads to more reliable detection and classification.

39th Conference on Neural Information Processing Systems (NeurIPS 2025).

Despite this potential, breast cancer analysis on chest NCCT poses several technical challenges. First, the absence of contrast enhancement in NCCT hampers lesion conspicuity against surrounding tissue. Second, lesion detection and malignancy classification serve complementary clinical purposes: the former focuses on localization, the latter supports diagnosis. Yet they are often treated as separate tasks, limiting the opportunity to share relevant features. Third, most existing approaches process each breast separately, overlooking the bilateral context that radiologists routinely use to detect asymmetries and subtle lesions. These limitations underscore the need for a unified framework that jointly models both tasks and captures long-range interactions between bilateral breast regions.

However, building such a unified framework remains challenging [60]. Segmentation architectures like UNet [51] and its variants rely on convolutional backbones with limited capacity for global context [30, 52], while vision transformers (ViTs) [1, 14] enable long-range modeling but suffer from high computational cost or low spatial resolution [21], compromising fine-grained lesion localization/segmentation. A promising direction lies in selective state space models (SSMs), such as Mamba [10, 17], which offer linear computational complexity with respect to sequence length, making it possible to use smaller patch sizes while preserving the ability to model long-range dependencies.

In this work, we propose Dual-Res Tandem Mamba-3D (DRT-M3D), a multitask framework that jointly performs segmentation-based lesion detection [1] and breast-level malignancy classification. Based on Mamba (S6) [10, 17], a selective state space model with linear complexity, DRT-M3D efficiently captures long-range dependencies across 3D volumes while preserving fine spatial granularity, which is crucial for detecting subtle lesions.

DRT-M3D introduces a dual-resolution architecture that separates voxel-level segmentation and breast-level classification into high-resolution and low-resolution paths, respectively. Mutual fusion modules align information across these two paths to improve representation quality without task interference. To model inter-breast context, we introduce a tandem input mechanism that processes bilateral breast volumes jointly, enabling cross-side interaction and the learning of asymmetry-aware representations, which are critical for detecting subtle signs of malignancy.

Experiments on three internal datasets and one external dataset demonstrate that DRT-M3D enables effective breast cancer analysis on chest NCCT scans, achieving state-of-the-art performance in both breast lesion segmentation and cancer classification tasks. Extensive ablation studies verify the effectiveness of each design choice, and we demonstrate that the tandem input strategy can also benefit vision transformer models when extended appropriately.

We summarize our contributions as follows:

- We propose a unified multi-task framework, DRT-M3D, for segmentation-based lesion detection and breast-level classification, enabling opportunistic breast cancer analysis on non-contrast chest CT scans.
- We design a dual-resolution architecture with mutual fusion to balance fine-grained spatial detail and global semantic context.
- We introduce a tandem input mechanism for bilateral modeling, enabling the network to leverage cross-side context as a structural prior.
- Evaluation on multi-institutional datasets shows that our model surpasses competitive baselines on both internal and external cohorts, demonstrating strong potential for clinical application.

## 2 Related Works

### 2.1 Breast Lesion Analysis in Medical Imaging

Breast lesion analysis has been widely studied across dedicated imaging modalities such as mammography [35, 53, 54], ultrasound [7, 23, 27, 65, 66], MRI [26, 71], and multi-modal scenarios [15, 28, 50], with deep learning models addressing tasks including lesion detection, segmentation, and malignancy classification. In contrast, research on breast cancer using non-contrast chest CT (NCCT) scans remains relatively underexplored [29, 33, 56]. U-Net variants dominate segmentation tasks

---

[1]Throughout this paper, *detection* refers to *segmentation-based lesion detection*, *i.e.*, identifying lesions through voxel-wise segmentation maps. We use *segmentation* and *detection* interchangeably in this context.

across modalities [3,37,68], while classification is typically performed via feature aggregation from segmentation backbones [39,72] or using standalone CNN/ViT-based classifiers [16,47].

Bilateral comparison is a routine practice in clinical mammography, and learning-based methods have explored dual-view fusion and symmetry-aware modeling [49,67]. A few methods incorporate handcrafted symmetry features or pairwise comparisons [5,12]. However, most approaches operate on 2D images and lack end-to-end bilateral modeling. In NCCT, where lesion visibility is often subtle due to low contrast, bilateral context is especially valuable yet remains underutilized, forming one of the key motivations of this study.

## 2.2 State Space Models for 3D Visual Data

State space models (SSM), such as structured state space sequence models (S4) [18–20], have emerged as linear-complexity alternatives to Transformers [59], alleviating the quadratic cost of attention for long-sequence processing. Mamba (S6) [10,17] extends S4 with a selective mechanism, and has proven effective in capturing long-range dependencies in visual data, making it a compelling alternative to Vision Transformers [14], especially in 3D tasks. VideoMamba [36] extends bidirectional Mamba layers from 2D [73] to 3D (video) data. VMamba [44] introduces a tailored SSM scanning strategy for 2D images, while Mamba-ND [38] generalizes into Mamba-3D blocks, simplifying volumetric modeling without complicating the internal SSMs. E-ViM³ [70] further demonstrates that pre-training Mamba-3D as a masked autoencoder (MAE) [22,58,63] boosts downstream task performance, echoing MAE's success in ViTs [14].

In medical image analysis, U-Mamba [46] integrates Mamba into UNet [51] backbone, though it demonstrates no clear benefits over conventional segmentation models [31]. SegMamba [64] introduces tri-orientated Mamba modules to better capture 3D context. Swin-UMamba [42,43] leverages VMamba [44] backbones pre-trained on ImageNet-1K [11] to improve performance on medical images, but remains limited to 2D analysis. EM-Net [6] combines Mamba with frequency-domain learning for multi-scale 3D features. More recently, Tri-Plane Mamba [62] incorporates SSMs into the Segment Anything Model (SAM) [32] for efficient interactive 3D segmentation.

Despite recent progress, existing methods are typically single-task, focusing solely on segmentation, without classification or bilateral context. In contrast, we integrate multi-task learning and bilateral-aware modeling into Mamba-3D, yielding a unified framework for breast cancer analysis on NCCT.

# 3 Methods

## 3.1 Overall Pipeline

Our pipeline for opportunistic breast cancer analysis on non-contrast chest CT (NCCT) scans is illustrated in Fig. 2. It takes the full NCCT 3D volume as input and produces both voxel-wise breast lesion segmentation and breast-level malignancy classification. The segmentation-based detection result highlights both benign and malignant lesions, guiding clinicians to focus on relevant areas. The classification result automatically identifies potentially malignant regions.

To focus on the breast regions within the entire chest CT, we introduce a Pre-Stage using a pre-trained segmentation network to perform coarse localization and cropping of each breast. The Main-Stage performs core analysis on the cropped breast regions, with all methods using consistent pre-processing and post-processing for fair evaluation. Implementation details are provided in Sec. C.

## 3.2 Dual-Res Mamba-3D

We first introduce the Dual-Res Mamba-3D (DR-M3D) network, which addresses the two interrelated yet distinct tasks: segmentation-based detection of breast lesions and malignancy classification on NCCT scans. DR-M3D operates independently on each breast region and also serves as a baseline for our study, in contrast to our full Dual-Res Tandem Mamba-3D (DRT-M3D) model.

**3D data embedding**   Given a 3D CT image $X \in \mathbb{R}^{1 \times D \times H \times W}$ (with 1 for one channel in grayscale CT, and $D, H, W$ for depth, height, and width), we apply a 3D embedding layer to transforms $X$ into the embedded feature map $Y \in \mathbb{R}^{C \times D_P \times H_P \times W_P}$. Each token in $Y$ with embedding dimension C

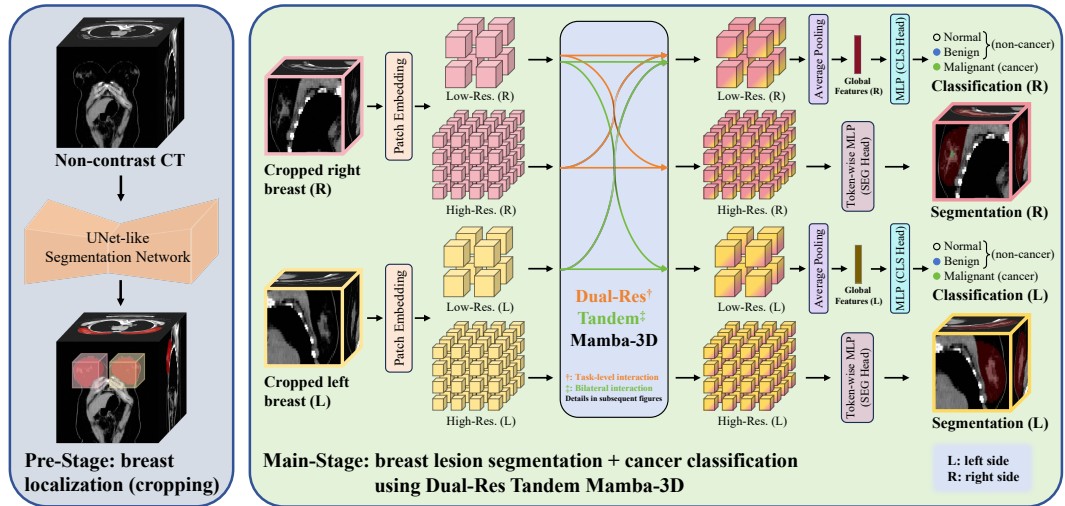

Figure 2: **The overall pipeline of the opportunistic breast cancer analysis approach.** The Pre-Stage employs a pre-trained segmentation network to perform coarse-grained localization and cropping of each breast region. The Main-Stage consists of the proposed Dual-Res Tandem Mamba-3D (DRT-M3D) network, which jointly performs segmentation-based detection of breast lesions and malignancy classification by leveraging dual-resolution features (intuitively represented by orange arrows) and facilitates cross-breast feature interaction between bilateral breast regions (green arrows). Please refer to Sec. 3.2, Sec. 3.3 and Fig. 3 for more details.

corresponds to a $p_d \times p_h \times p_w$ patch in the original image, where $p_d = \frac{D}{D_p}$, $p_h = \frac{H}{H_p}$, and $p_w = \frac{W}{W_p}$. We independently apply amplitude-learnable sinusoidal embeddings along each dimension and add them to equally split segments of the embedding space to help preserve spatial structure.

**Dual-resolution design with mutual fusion** To support both fine-grained lesion segmentation and breast-level malignancy classification, DR-M3D adopts a dual-resolution architecture. Specifically, the embedded feature map $Y \in \mathbb{R}^{C \times D_P \times H_P \times W_P}$ is further downsampled via 3D MaxPooling to generate a lower-resolution feature map $Z \in \mathbb{R}^{C' \times D'_P \times H'_P \times W'_P}$. Here, $C'$ denotes the increased embedding dimension for a more powerful representation for the low-resolution path, and the downsampling factors are defined as $p'_d = \frac{D_P}{D'_P}$, $p'_h = \frac{H_P}{H'_P}$, and $p'_w = \frac{W_P}{W'_P}$. Thus, $Y$ and $Z$ are used as the initial inputs to the high-resolution (HR) and low-resolution (LR) paths, respectively, denoted as $Y^0$ and $Z^0$ in the following text.

As the DRT-M3D block shown in Fig. 3 (a), each DR-M3D block also contains an HR and an LR Mamba-3D sub-block, forming a dual-path structure. From a black-box view, the $i$-th DR-M3D block defines the mapping $(Y^{i+1}, Z^{i+1}) = \text{DR-M3D}^i(Y^i, Z^i)$ while preserving the spatial resolutions of both paths. The HR path retains spatial details for voxel-level segmentation, whereas the LR path, operating on shorter sequences, captures long-range dependencies crucial for classification.

To facilitate cross-path information exchange, each DR-M3D block integrates a mutual fusion mechanism. Specifically, after the HR Mamba-3D sub-block processes $Y^i$, the output $\widetilde{Y}^{i+1}$ is downsampled and projected to match the LR resolution and channel dimension, allowing residual fusion with the LR path. Conversely, the LR output $\widetilde{Z}^{i+1}$ is upsampled and projected to match the HR shape before being added to the input of the next HR block.

Formally, the internal update process within the $i$-th DR-M3D block can be expressed as:

$$\begin{cases} \widetilde{Y}^i = Y^i, & \widetilde{Z}^i = Z^i + f^i_{\text{down}}(\widetilde{Y}^{i+1}) \\ Y^{i+1} = \widetilde{Y}^{i+1} + f^i_{\text{up}}(\widetilde{Z}^{i+1}), & Z^{i+1} = \widetilde{Z}^{i+1} \end{cases} \quad (1)$$

where

$$\begin{cases} \widetilde{Y}^{i+1} = \text{Mamba-3D}(\widetilde{Y}^i) \\ \widetilde{Z}^{i+1} = \text{Mamba-3D}(\widetilde{Z}^i) \end{cases} \quad (2)$$

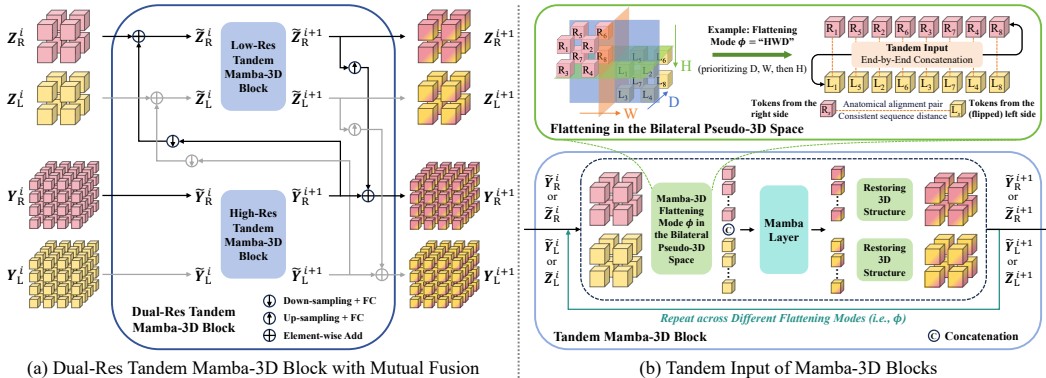

(a) Dual-Res Tandem Mamba-3D Block with Mutual Fusion    (b) Tandem Input of Mamba-3D Blocks

Figure 3: **Dual-Res Tandem Mamba-3D block.** (a) The high-resolution (HR) path captures fine-grained local features, while the low-resolution (LR) path focuses on global context. The upsampled LR output is fused back to the HR path, facilitating joint feature alignment. (b) The Tandem Mamba-3D block flattens both 3D inputs to 1D with specific dimension-priority flattening modes and concatenates them to form the Tandem Input. This prevents mixing of bilateral information while enabling cross-side feature interaction during selective scanning.

and the downsampling and upsampling transformations are defined as:

$$
\begin{cases}
f_{\text{down}}^i(\widetilde{\boldsymbol{Y}}^{i+1}) = \text{Linear}_{\text{C}\to\text{C}'}\big[\text{3D-MaxPooling}(\widetilde{\boldsymbol{Y}}^{i+1})\big] \\
f_{\text{up}}^i(\widetilde{\boldsymbol{Z}}^{i+1}) = \text{Linear}_{\text{C}'\to\text{C}}\big[\text{3D-Interpolation}(\widetilde{\boldsymbol{Z}}^{i+1})\big]
\end{cases}
\tag{3}
$$

Our dual-resolution architecture assigns tasks to different paths, while residual coupling via mutual fusion encourages joint feature alignment. This setup can be seen as a form of soft parameter sharing [9] under a multi-task optimization framework, also emulating how radiologists integrate local and global cues in clinical reading.

### 3.3 Tandem Input for Mamba-3D Blocks

Building on the DR-M3D structure introduced in Sec. 3.2, we further seek to exploit the inherent correlation between bilateral breasts, aiming to mitigate the limited feature richness of NCCT images. To achieve this, we propose the tandem input mechanism as illustrated in Fig. 3 (b), which transforms each sub-block into a Tandem Mamba-3D block, forming the complete DRT-M3D architecture.

Specifically, we organize the tokens from the left and right breasts into a bilateral pseudo-3D space. The right breast occupies the coordinate range from $(1, 1, 1)$ to $(D_R, H_R, W_R)$, while the left is positioned from $(D_R + 1, H_R + 1, W_R + 1)$ to $(D_R + D_L, H_R + H_L, W_R + W_L)$. For anatomical alignment, the left breast is pre-flipped along the X-axis, so both regions share the same orientation and are juxtaposed within the extended space, without assuming direct connectivity along any particular axis. By flattening the pseudo-3D volume while skipping unoccupied regions, we obtain the tandem input sequence as shown in Fig. 3 (b), which is equivalent to flattening the left and right breast regions separately with the same mode and then concatenate them end-to-end. The internal different 3D flattening modes and scanning processes follow the Mamba-3D strategy [38, 70], as detailed in Sec. A and illustrated in Fig. 7.

The tandem input design not only ensures that Mamba-3D scans all voxels from one breast before moving to the other, but also guarantees a roughly fixed distance between anatomically symmetric locations in the sequence, allowing the model to better capture cross-breast relationships. The LR path further enhances cross-side modeling by offering a shorter sequence and higher embedding dimensionality. Moreover, the prioritized scanning strategy across different axes aligns with the multi-orientation analysis and comprehensive assessment performed in real clinical evaluation.

### 3.4 Training Procedure

**Efficient self-supervised pre-training** We adopt the efficient self-supervised pre-training strategy for Mamba-3D blocks proposed in [70], which improves downstream performance and accelerates convergence without extra data. We set the size of jointly masked regions in the HR path to match the downsampling factors $(p_d', p_h', p_w')$ from HR to LR, with a masking ratio $p_{\text{mask}}$. This ensures

that each LR token is computed only from unmasked HR tokens, avoiding information leakage and allowing for the removal of masked tokens during pre-training.

To construct the masked autoencoder [22, 58, 70], we append $M$ additional DRT-M3D blocks as the decoder following the backbone network. The pretext task is to reconstruct the masked tokens from both the HR and LR outputs via token-to-patch linear projections. The pretraining objective is optimized using a mean squared error (MSE) loss, formulated as:

$$\mathcal{L}_{\text{PRE}} = \frac{1}{|V_{\text{masked}}|} \sum_{j=1}^{|V_{\text{masked}}|} \left[ \left( \left[ \boldsymbol{X}_{\text{Y-pred}} \right]_j - \left[ \boldsymbol{X} \right]_j \right)^2 + \left( \left[ \boldsymbol{X}_{\text{Z-pred}} \right]_j - \left[ \boldsymbol{X} \right]_j \right)^2 \right] \tag{4}$$

where $V_{\text{masked}}$ denotes the set of masked voxels.

**Joint segmentation and classification fine-tuning**    We fine-tune the network jointly for voxel-wise lesion segmentation and breast-level malignancy classification, while the network structure enforces partial task decoupling and soft feature interaction through the dual-resolution paths.

For the segmentation task, supervision is applied to the HR path's output using a hybrid loss composed of the Dice loss [48] and voxel-level cross-entropy (CE) loss:

$$\mathcal{L}_{\text{SEG}} = \text{Dice}\left( \boldsymbol{M}_{\text{Y-pred}}, \boldsymbol{M}_{\text{gt}} \right) + \frac{1}{|V|} \sum_{j=1}^{|V|} \text{CE}\left( \left[ \boldsymbol{M}_{\text{Y-pred}} \right]_j - \left[ \boldsymbol{M}_{\text{gt}} \right]_j \right)^2 \tag{5}$$

where $\boldsymbol{M}_{\text{Y-pred}}$ is the segmentation prediction obtained from the HR output $\boldsymbol{Y}^{N-1}$ via an MLP, $\boldsymbol{M}_{\text{gt}}$ is the ground truth mask, and V denotes the set of all voxels in the cropped 3D-image.

For the classification task, supervision is applied to the LR path's output using standard image-level cross-entropy loss:

$$\mathcal{L}_{\text{CLS}} = \text{CE}(\boldsymbol{y}_{\text{Z-pred}}, \boldsymbol{y}_{\text{gt}}) \tag{6}$$

where $\boldsymbol{y}_{\text{Z-pred}}$ is the predicted malignancy score obtained by applying global average pooling and an MLP to the LR output $\boldsymbol{Z}^{N-1}$, and $\boldsymbol{y}_{\text{gt}}$ is the ground-truth breast-level malignancy label.

The overall fine-tuning loss is the weighted sum of the two task losses:

$$\mathcal{L}_{\text{FT}} = \mathcal{L}_{\text{SEG}} + \lambda \mathcal{L}_{\text{CLS}} \tag{7}$$

where $\lambda$ balances the relative importance of the two tasks. We empirically set $\lambda = 0.1$, reflecting the relative simplicity and higher overfitting risk of the classification task.

## 4   Experiments

### 4.1   Experimental Setup

**Datasets**    Following the medical image analysis paradigm, we use both internal and external datasets. The internal datasets include training and testing splits, while the external set is reserved for testing only, providing a distribution shift that enables a rigorous assessment of generalization.

The internal data comprises three cohorts collected from separate institutions, referred to as Inst. 1-3, with training and testing splits contain {341 / 315, 239 / 921, 141 / 113} and {82 / 78, 102 / 296, 34 / 28} *cancerous / non-cancerous* breast cases, respectively. The external dataset contains 214 breast samples as {105 / 109} *cancerous / non-cancerous*.

For all datasets in the Main-Stage, bilateral breasts from the same patient are always assigned to the same split to avoid data leakage. Each sample consists of a cropped NCCT image of the breast region, segmentation masks for the breast and lesions (if present), and a binary cancer label. More details regarding the datasets and the Pre-Stage process can be found in Sec. B.

**Implementation**    Our model is implemented in PyTorch and trained on up to two NVIDIA A100 (80GB) GPUs. The training process consists of 500 epochs for self-supervised pre-training and 50 epochs for downstream fine-tuning, optimized using AdamW [45]. The base learning rate is

Table 1: **Quantitative results of the internal evaluation across three datasets.** The best and second-best results are **bolded** and underlined, respectively. (H.r.: Hit-Rate, F.s.: FROC-Score, Spec.: Specificity, Sens.: Sensitivity)

| Method | Inst. 1 (Spec. = 0.9615) | | | | | Inst. 2 (Spec. = 0.9595) | | | | | Inst. 3 (Spec. = 0.9643) | | | | |
|---|---|---|---|---|---|---|---|---|---|---|---|---|---|---|---|
| | Dice | H.r. | F.s. | Sens. | AUC | Dice | H.r. | F.s. | Sens. | AUC | Dice | H.r. | F.s. | Sens. | AUC |
| nnUNet [30] | 0.6174 | **0.9390** | 0.8338 | 0.8049 | 0.9558 | 0.6413 | 0.9099 | 0.8806 | 0.7647 | 0.9501 | 0.5980 | 0.8611 | 0.8264 | 0.7353 | 0.9170 |
| nnUNet-SEG | 0.6231 | 0.9268 | 0.8760 | - | - | 0.6548 | 0.9099 | 0.8694 | - | - | 0.5862 | 0.8333 | 0.8194 | - | - |
| nnUNet-CLS | - | - | - | 0.7927 | 0.9622 | - | - | - | 0.7549 | 0.9540 | - | - | - | 0.7353 | 0.9275 |
| VNet [48] | 0.5623 | 0.9146 | 0.8333 | 0.7683 | 0.9500 | 0.5821 | 0.8649 | 0.8198 | 0.7451 | 0.9418 | 0.5116 | 0.8056 | 0.7431 | 0.7353 | 0.8797 |
| swinUNETR [24] | 0.5647 | 0.9146 | 0.8043 | 0.8171 | 0.9432 | 0.6009 | 0.9009 | 0.8423 | 0.7353 | 0.9423 | 0.5465 | 0.8333 | 0.7764 | 0.7059 | 0.8981 |
| nnFormer [69] | 0.5601 | 0.9268 | 0.8130 | 0.6707 | 0.9595 | 0.6122 | 0.9099 | 0.8709 | 0.7549 | 0.9444 | 0.5298 | 0.8333 | 0.7709 | 0.7353 | 0.9307 |
| 3D UX-Net [34] | 0.5669 | 0.9268 | 0.7811 | 0.7683 | 0.9361 | 0.6015 | 0.9279 | 0.8288 | 0.7353 | 0.9382 | 0.5213 | 0.8333 | 0.7327 | 0.7647 | 0.9391 |
| MedNeXt [52] | 0.6175 | 0.9146 | 0.8719 | 0.7683 | 0.9580 | 0.6292 | **0.9369** | 0.8919 | 0.6863 | 0.9510 | 0.5604 | 0.8333 | 0.7848 | 0.7647 | 0.9076 |
| U-Mamba-Bot [46] | 0.6042 | 0.8659 | 0.8552 | 0.8049 | 0.9578 | 0.5989 | 0.7838 | 0.7928 | 0.7451 | 0.9596 | 0.5361 | 0.7500 | 0.7500 | 0.7941 | 0.9391 |
| U-Mamba-Enc [46] | 0.5708 | 0.8171 | 0.8171 | 0.7805 | 0.9548 | 0.5671 | 0.7838 | 0.7928 | 0.7451 | 0.9561 | 0.4901 | 0.6944 | 0.6944 | 0.7059 | 0.8792 |
| EM-Net [6] | 0.5803 | 0.9268 | 0.8059 | 0.7439 | 0.9467 | 0.6203 | 0.9189 | 0.8587 | 0.7549 | 0.9540 | 0.5461 | 0.8056 | 0.7361 | 0.6765 | 0.9160 |
| SegMamba [64] | 0.5852 | 0.9146 | 0.8509 | 0.7805 | 0.9555 | 0.6250 | 0.8919 | 0.8919 | 0.8039 | 0.9479 | 0.5109 | 0.7500 | 0.7152 | 0.7353 | 0.9074 |
| Sun et al. 2025 [56] | 0.6061 | 0.9268 | 0.8613 | 0.7561 | 0.9457 | 0.6317 | 0.9099 | 0.8649 | 0.7647 | 0.9437 | 0.5859 | 0.8889 | 0.7778 | 0.7059 | 0.9144 |
| **DR-M3D (ours)** | 0.6553 | 0.9309 | **0.9045** | **0.8780** | 0.9725 | 0.6744 | 0.9339 | **0.9234** | 0.7549 | 0.9686 | 0.6003 | 0.8704 | 0.8264 | 0.7505 | 0.9226 |
| | ±0.0083 | ±0.0070 | ±0.0064 | ±0.0122 | ±0.0063 | ±0.0075 | ±0.0104 | ±0.0181 | ±0.0098 | ±0.0016 | ±0.0078 | ±0.0161 | ±0.0184 | ±0.0245 | ±0.0097 |
| **DRT-M3D (ours)** | **0.6608** | 0.9349 | **0.9258** | **0.9471** | **0.9909** | **0.6750** | 0.9279 | 0.9212 | **0.8366** | **0.9743** | **0.6056** | 0.8796 | **0.8518** | **0.8726** | **0.9471** |
| | ±0.0018 | ±0.0070 | ±0.0063 | ±0.0186 | ±0.0015 | ±0.0022 | ±0.0090 | ±0.0060 | ±0.0299 | ±0.0020 | ±0.0148 | ±0.0161 | ±0.0080 | ±0.0170 | ±0.0044 |

set to $1e-3$, with a linear warm-up phase followed by cosine annealing. Detailed experimental configurations and hyper-parameters are provided in Sec. C.

For baselines not compatible with MAE pre-training (mainly CNN-based models), we use supervised pre-training on segmentation followed by joint fine-tuning on segmentation and classification, which achieves better performance and enables a fairer comparison. For architectures that support MAE pre-training (*i.e.*, Transformer or Mamba-based models), we apply MAE pre-training (without extra data) and fine-tune with the same protocol as our method.

**Evaluation metrics**  For segmentation-based detection, we evaluate performance using the Dice Similarity Coefficient (Dice), Hit-Rate (H.r.), and FROC-Score (F.s.) [41] derived from the Free-Response Receiver Operating Characteristic (FROC) curve [2]. Among these, Dice measures voxel-level segmentation quality, while Hit-Rate and FROC-Score assess lesion-level sensitivity and the ability to control false positives. For classification, we report sensitivity (Sens.) at a fixed high specificity (Spec.) threshold for different methods and the Area Under the Receiver Operating Characteristic Curve (AUC). Further details on evaluation metrics are available in Sec. D.

## 4.2  Main Results

**Quantitative comparisons on internal datasets** Tab. 1 presents quantitative results on three internal datasets, comparing our method against strong baselines and state-of-the-art approaches. For UNet-based methods, we incorporate multi-scale feature fusion [40, 72] for classification. To isolate the effect of task interaction, we include single-task nnUNet variants: nnUNet-SEG (segmentation-only) and nnUNet-CLS (classification-only). Our method, DRT-M3D, achieves leading performance across most metrics on all three datasets. In particular, it consistently improves segmentation-based detection by reducing false positives (as reflected in the FROC-score), and boosts cancer classification performance in terms of Sensitivity and AUC. We also note that our baseline DR-M3D model performs competitively, further validating the effectiveness of our dual-resolution design.

**Quantitative comparisons on the external dataset** The external dataset from a distinct institution (Inst. 4) poses a significant distribution shift, serving as a

Table 2: **Quantitative results of the external evaluation.** The best and second-best results are **bolded** and underlined, respectively. (H.r.: Hit-Rate, F.s.: FROC-Score, Spec.: Specificity, Sens.: Sensitivity)

| Method | External (Spec. = 0.9083) | | | | |
|---|---|---|---|---|---|
| | Dice | H.r. | F.s. | Sens. | AUC |
| nnUNet [30] | 0.5668 | 0.8889 | 0.8173 | 0.6286 | 0.8889 |
| nnUNet-SEG | 0.5776 | 0.8632 | 0.8173 | - | - |
| nnUNet-CLS | - | - | - | 0.7238 | 0.9156 |
| VNet [48] | 0.5059 | 0.8120 | 0.7272 | 0.6571 | 0.8790 |
| swinUNETR [24] | 0.5364 | 0.8803 | 0.7639 | 0.6476 | 0.8818 |
| nnFormer [69] | 0.5270 | 0.8889 | 0.7568 | 0.6190 | 0.8806 |
| 3D UX-Net [34] | 0.5217 | 0.8547 | 0.7589 | 0.6095 | 0.8777 |
| MedNeXt [52] | 0.5767 | 0.8974 | 0.8120 | 0.6571 | 0.8945 |
| U-Mamba-Bot [46] | 0.5415 | 0.7863 | 0.7799 | 0.7238 | 0.8974 |
| U-Mamba-Enc [46] | 0.5180 | 0.7436 | 0.7308 | 0.6095 | 0.8893 |
| EM-Net [6] | 0.5489 | 0.8547 | 0.7618 | 0.6000 | 0.8838 |
| SegMamba [64] | 0.5388 | 0.8462 | 0.7639 | 0.6381 | 0.9083 |
| Sun et al. 2025 [56] | 0.5624 | 0.8974 | 0.8246 | 0.6286 | 0.8827 |
| **DR-M3D (ours)** | 0.5909 | 0.8974 | 0.8632 | 0.7207 | 0.9082 |
| | ±0.0038 | ±0.0041 | ±0.0064 | ±0.0092 | ±0.0091 |
| **DRT-M3D (ours)** | **0.5948** | **0.9117** | **0.8766** | **0.8762** | **0.9371** |
| | ±0.0027 | ±0.0082 | ±0.0057 | ±0.0080 | ±0.0046 |

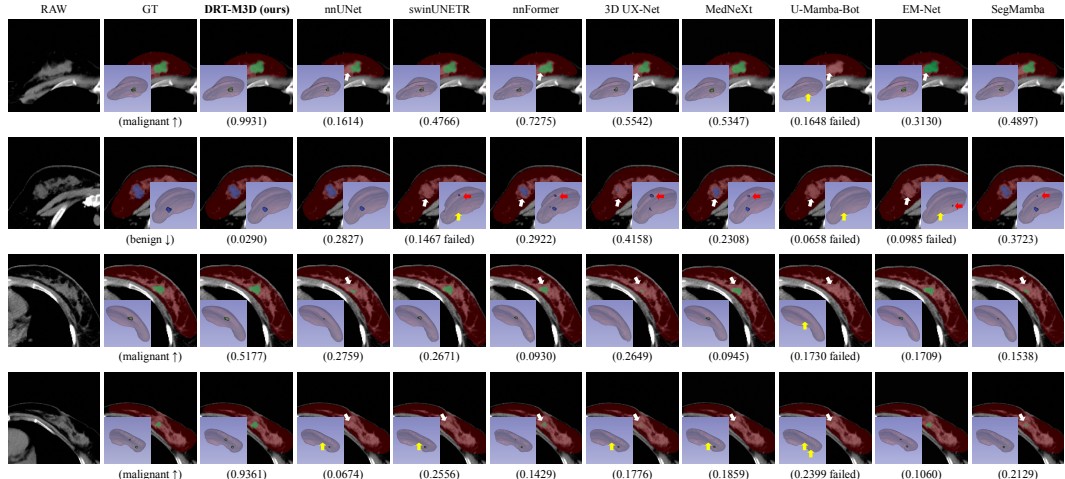

Figure 4: **Qualitative comparison of DRT-M3D with competing methods.** Examples from internal and external test sets are shown, including a representative 2D slice and a 3D view for each case. Red, green, and blue masks represent breast regions, malignant, and benign lesions, respectively. White, yellow, and red arrows mark pronounced segmentation errors in the slice, missed lesions, and segmentation of non-existent lesions, respectively. Predicted malignancy scores are shown for each case (higher for malignant, lower for benign). "Failed" denotes cases where none of the true lesions were localized (*i.e.*, a non-hit under the Hit-Rate metric).

Table 3: **Ablation study on the designs of the proposed DRT-M3D network.** DR, MF, and TI denote the use of Dual-Resolution, Mutual Fusion, and Tandem Input, respectively. (H.r.: Hit-Rate, F.s.: FROC-Score, Spec.: Specificity, Sens.: Sensitivity)

| Variants | | | | Internal (Spec. = 0.9627) | | | | | External (Spec. = 0.9083) | | | | |
|---|---|---|---|---|---|---|---|---|---|---|---|---|---|
| DR | MF | TI | | Dice | H.r. | F.s. | Sens. | AUC | Dice | H.r. | F.s. | Sens. | AUC |
| | | | Vanilla M3D | 0.6334 | 0.8908 | 0.8854 | 0.7936 | 0.9637 | 0.5788 | 0.8666 | 0.8359 | 0.7100 | 0.9053 |
| ✓ | | | | 0.6449 | 0.9170 | 0.8879 | 0.7982 | 0.9574 | 0.5848 | 0.8879 | 0.8509 | 0.6620 | 0.8940 |
| ✓ | ✓ | | DR-M3D | 0.6564 | 0.9228 | 0.9027 | 0.8073 | 0.9690 | 0.5909 | 0.8974 | 0.8632 | 0.7207 | 0.9082 |
| | | ✓ | | 0.6388 | 0.8952 | 0.8876 | 0.8211 | 0.9625 | 0.5898 | 0.8880 | 0.8565 | 0.8522 | 0.9184 |
| ✓ | | ✓ | | 0.6491 | 0.9214 | 0.9072 | 0.8303 | 0.9719 | 0.5925 | 0.9022 | 0.8698 | 0.8762 | 0.9264 |
| ✓ | ✓ | ✓ | DRT-M3D | **0.6590** | **0.9229** | **0.9145** | **0.8578** | **0.9768** | **0.5948** | **0.9117** | **0.8766** | **0.8762** | **0.9371** |

strong generalization benchmark. As shown in Tab. 2, DRT-M3D maintains leading performance across all metrics, outperforming all competing methods. The variant without tandem input (DR-M3D) also generalizes well, especially for segmentation, further supporting the robustness of the overall architecture.

For both DR-M3D and DRT-M3D, we report the mean and standard deviation of each metric over five runs with different random seeds, as reported in Tab. 1 and Tab. 2.

**Visualization results for qualitative comparison**
Fig. 4 shows representative lesion detection and classification outputs. DRT-M3D delivers accurate localization and better differentiation between cancerous and non-cancerous cases. Competing methods more frequently miss lesions or misclassify malignant breasts.

### 4.3 Ablation Studies

**Ablation on architectural designs** We evaluate the impact of key components in DRT-M3D through ablation studies on Dual-Resolution (DR, w/o means using the HR path for both tasks), Mutual Fusion (MF), and Tandem Input (TI). The results in Tab. 3 demonstrate that the DR design enhances segmentation, while TI improves breast-level cancer classifica-

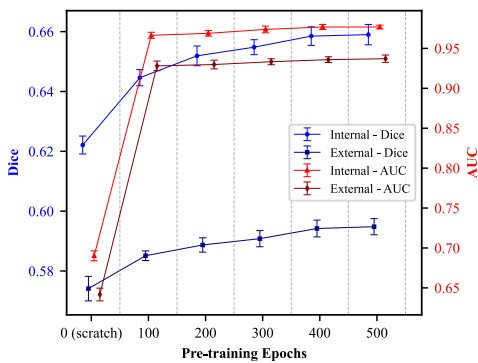

Figure 5: **Ablation study on the training strategy.** Error bars on the curves are obtained by repeating the experiments with five different random seeds.

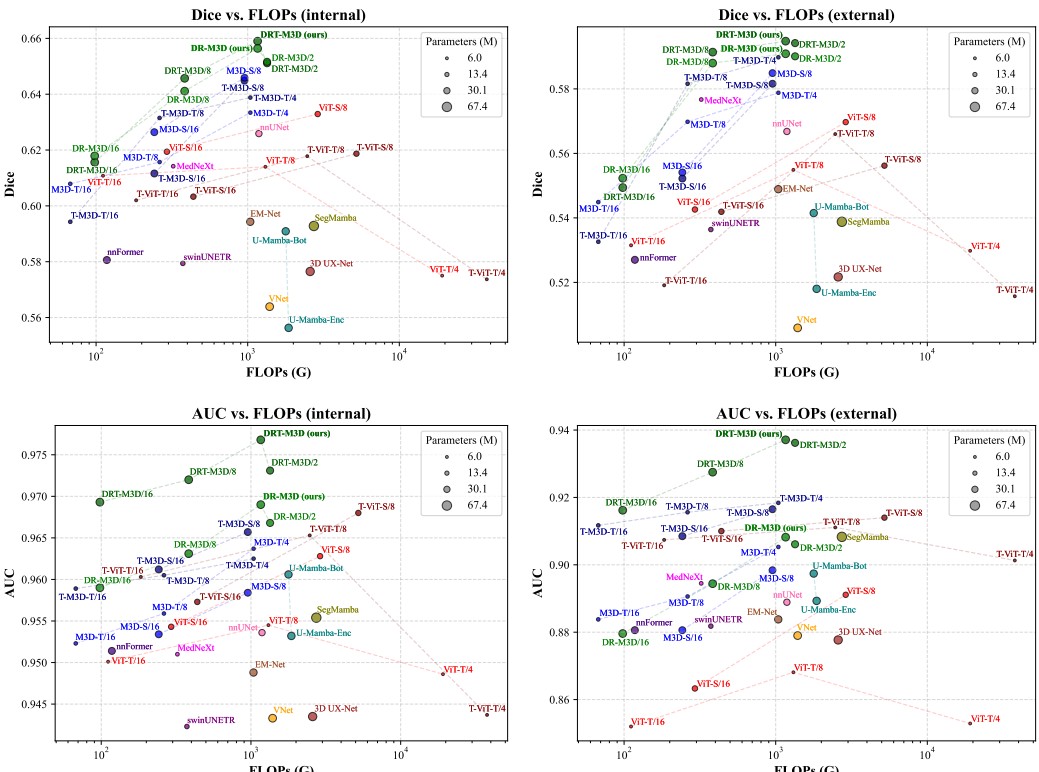

Figure 6: **Comparison of Dice and AUC vs. FLOPs on internal and external datasets.** Each FLOPs value is computed per bilateral input (or two unilateral inputs). ViT: vanilla Vision Transformer, M3D: vanilla Mamba-3D, T-~: tandem (bilateral) input form, ~/p: patch size of (2, p, p).

tion. Additionally, the MF mechanism facilitates information sharing between the Dual-Res paths, boosting performance across all metrics. For Tandem Input specifically, we further compare several variants in Sec. E.

**Ablation on the training strategy**    MAE pre-training has been shown to benefit Mamba-3D on high-level semantic tasks [70]. Since our study also involves voxel-level segmentation, we further validate its effectiveness in this context. To fairly compare different pre-training durations, we adjust the fine-tuning epochs for full convergence. As shown in Fig. 5, the two-stage training strategy with adequate self-supervised pre-training also proves effective for the multi-task setting of segmentation and classification in this study.

**Patch size tuning**    We further validate through experiments (see Fig. 6) that using a relatively smaller patch size in height ($p_h$) and width ($p_w$) of CT slices, such as (2, 4, 4) in our models, substantially improves performance. Due to the model's underlying linear computation design, this gain comes with minimal computational overhead. However, further reducing the patch size (*e.g.*, to (2, 2, 2)) leads to stagnation or even a drop in performance, presumably due to increased token fragmentation and reduced contextual capacity. Conversely, variants with larger patch sizes can significantly reduce computational costs while still outperforming competing methods. Detailed configurations of each variant are provided in Appendix Tab. 5 and Tab. 6. For experiments regarding the patch size in the depth dimension ($p_d$), please refer to Sec. E.

**Bilateral input on vision transformers**    To further evaluate the Mamba-3D (M3D) structure and tandem input concept, we integrate bilateral input into the commonly used ViT [14] for comparison. We adopt ViT-T and ViT-S as two different scales, and denote their bilateral variants as T-ViT. In T-ViT, left-side and right-side encodings are added to the 3D positional encoding to distinguish patches. All ViT models follow the two-stage training pipeline as ours. As shown in Fig. 6, bilateral input improves cancer classification in ViT models but still fall short compared to our M3D-based models, particularly for ViT/4 with the same patch size (2, 4, 4) as DRT-M3D. The performance gap

is primarily due to the excessive number of tokens resulting from small patch sizes and bilateral input, which leads to peaked and less expressive attention distributions and thus hampers the learning of patch relationships [57]. For larger patch sizes, ViT's performance remains clearly inferior, especially in segmentation. Although larger patches reduce token count, they further decrease spatial resolution, and the lack of intrinsic local inductive bias in ViT prevents effective modeling of local structures.

## 5 Conclusion

In this paper, we propose DRT-M3D, a dual-resolution network for joint segmentation-based breast lesion detection and cancer classification on non-contrast chest CT scans. By disentangling the two tasks into resolution-specific pathways, DRT-M3D enables complementary learning between segmentation and classification, while tandem bilateral inputs enhance contextual understanding of subtle features across bilateral breasts. Experiments on multi-institutional datasets demonstrate consistent improvements over strong baselines, underscoring the clinical potential of DRT-M3D for robust and generalizable opportunistic breast cancer analysis.

**Limitations and future work** This study only focuses on breast cancer analysis using NCCT scans, but the idea of bilateral organs and the design of DRT-M3D may be applied to other organs such as lungs and kidneys. In addition, the data currently in use all come from one country and have not been extended to a broader population. We have already started to collect more diverse data from a wider range of regions for future research.

## Acknowledgments and Disclosure of Funding

This work was supported in part by the National Natural Science Foundation of China (grant 92354307), the National Key Research and Development Program of China (grant 2024YFF0729202), and the Strategic Priority Research Program of the Chinese Academy of Sciences (grant XDA0460305). This work was also supported by Alibaba Group through Alibaba Research Intern Program.

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

# A  Preliminaries on Mamba-3D Blocks

## A.1  Selective State Space Models

Selective state space models (S6), which are represented by Mamba [17], are essentially derived from the vanilla state space models (SSM) based on continuous linear time-invariant (LTI) systems.

SSMs utilize N-dimensional latent state vector $\mathbf{h}(t) \in \mathbb{R}^{\mathrm{N}}$ to model the transformation from a one-dimensional continuous input signal to its corresponding output $x(t) \in \mathbb{R} \rightarrow y(t) \in \mathbb{R}$:

$$
\begin{aligned}
\mathbf{h}'(t) &= \mathbf{A}\mathbf{h}(t) + \mathbf{B}x(t) \\
y(t) &= \mathbf{C}\mathbf{h}(t) + \mathbf{D}x(t)
\end{aligned}
\tag{8}
$$

where $\mathbf{A} \in \mathbb{R}^{\mathrm{N \times N}}$ is the "state matrix", $\mathbf{B} \in \mathbb{R}^{\mathrm{N \times 1}}$ is the "input matrix", $\mathbf{C} \in \mathbb{R}^{\mathrm{1 \times N}}$ is the "output matrix", and $\mathbf{D} \in \mathbb{R}^{\mathrm{1 \times 1}}$ is the "feed-through matrix". These are system matrices governing the evolution and output of the system.

To make these models applicable in deep learning settings with discrete inputs, structured state space models (S4) [18–20] discretize these equations using numerical methods such as the zero-order hold (ZOH) [55] approach, yielding the following discretization:

$$
\begin{aligned}
\overline{\mathbf{A}} &= \exp(\Delta \mathbf{A}) \\
\overline{\mathbf{B}} &= (\Delta \mathbf{A})^{-1} \left[\exp(\Delta \mathbf{A}) - \mathbf{I}\right] \cdot \Delta \mathbf{B} \approx \Delta \mathbf{B}
\end{aligned}
\tag{9}
$$

where $\Delta$ denotes the discretization step size. The larger the step size $\Delta$, the faster the hidden state $\mathbf{h}$ changes, and the greater the impact of the current token $x$ on the system.

In Mamba, $\mathbf{A}$ must be a diagonal matrix. Consequently, when entries of $\Delta \mathbf{A}$ are sufficiently small, the approximation $(\Delta \mathbf{A})^{-1} \left[\exp(\Delta \mathbf{A}) - \mathbf{I}\right] \approx \mathbf{I}$ holds true. This leads to the discrete-time S4 model for sequential input and output, as $x_t \in \mathbb{R} \rightarrow y_t \in \mathbb{R}$:

$$
\begin{aligned}
\mathbf{h}_t &= \overline{\mathbf{A}}\mathbf{h}_{t-1} + \overline{\mathbf{B}}x_t \approx \exp(\Delta \mathbf{A})\mathbf{h}_{t-1} + \Delta \mathbf{B}x_t \\
y_t &= \mathbf{C}\mathbf{h}_t + \mathbf{D}x_t
\end{aligned}
\tag{10}
$$

For multi-channel inputs (*i.e.*, $\mathbf{x}_t \in \mathbb{R}^{\mathrm{C}}$) and the corresponding multi-channel outputs, the above operations are applied independently to each channel.

Building upon S4, the S6 structure enhances the model's expressiveness by making the parameters $\overline{\mathbf{A}}$, $\overline{\mathbf{B}}$, $\mathbf{C}$ dynamically input-dependent across all channels of $\mathbf{x}_t \in \mathbb{R}^{\mathrm{C}}$, which is known as the selection mechanism:

$$
\begin{aligned}
\mathbf{B}_t &= \mathrm{Linear}_{\mathrm{C \rightarrow N}}(\mathbf{x}_t) \\
\mathbf{C}_t &= \mathrm{Linear}_{\mathrm{C \rightarrow N}}(\mathbf{x}_t) \\
\mathbf{\Delta}_t &= \mathrm{softplus}\left[\mathrm{Linear}_{\mathrm{C \rightarrow C}}(\mathbf{x}_t)\right]
\end{aligned}
\tag{11}
$$

With the discretization in Eq. (9), $\overline{\mathbf{A}}$, $\overline{\mathbf{B}}$, $\mathbf{C}$ are no longer fixed parameters in the inference phase, allowing the model to adapt to varying input signals and provides greater modeling capabilities than the original SSMs. Overall, S6 can be expressed as:

$$
\begin{aligned}
\mathbf{h}_t &= \exp(\mathbf{\Delta}_t \mathbf{A})\mathbf{h}_{t-1} + \mathbf{\Delta}_t \mathbf{B}_t x_t \\
y_t &= \mathbf{C}_t \mathbf{h}_t + \mathbf{D}x_t
\end{aligned}
\tag{12}
$$

The input-dependent nature of S6 (Mamba) makes it particularly suitable for processing patchified visual data, where different spatial regions may require distinct dynamic responses, similar to words in natural language processing. The exponential transition term captures long-range dependencies, while the learned $\Delta_t$ modulates temporal sensitivity, making Mamba capable of adapting to both local and global features.

## A.2  Mamba-3D Blocks

Unlike most visual Mamba works [36,44,73] that modify the internal structure of S6 blocks to support multi-directional scanning, Mamba-3D [38,70] retains the original architecture of the Mamba block,

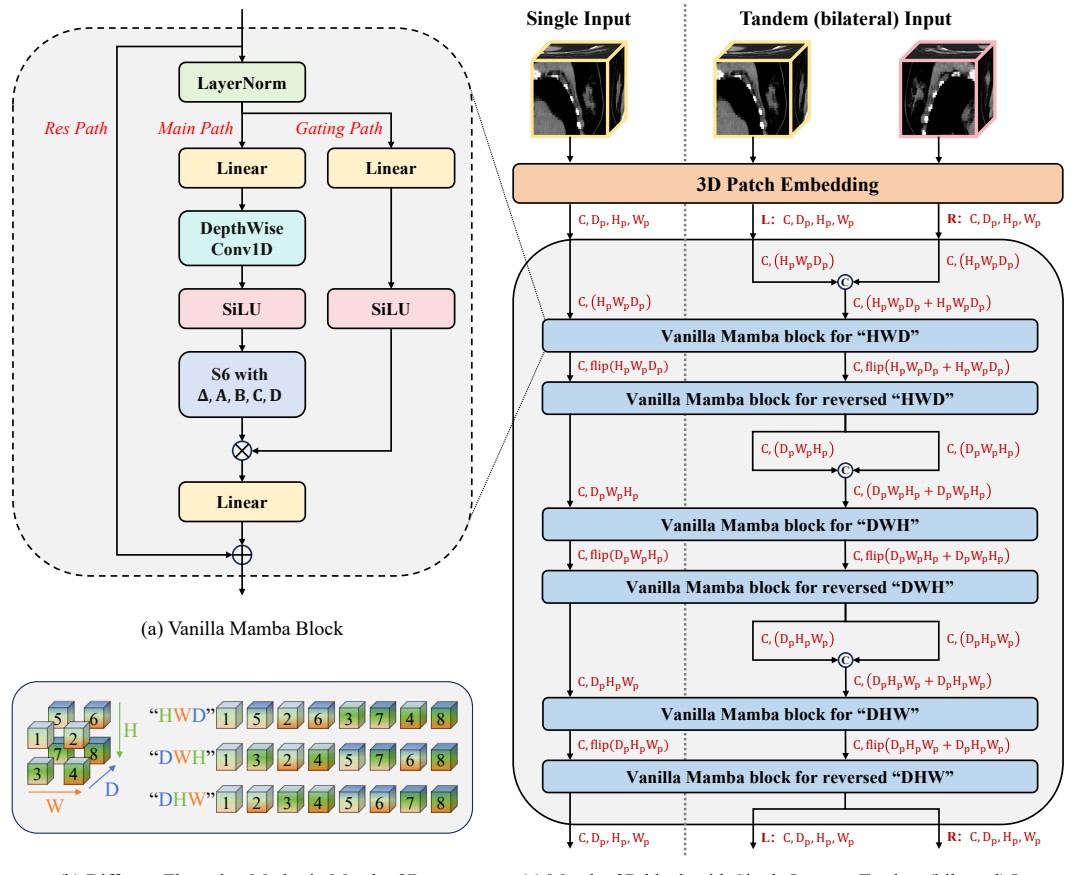

Figure 7: **The internal structure of the proposed Tandem Mamba-3D block.** (a) Vanilla Mamba block with selective state space module (S6). (b) Different sequence permutations used for the serial scanning processes in Mamba-3D. (c) Data flow inside one Mamba-3D block in the cases of Single Input and Tandem (bilateral) Input.

as depicted in Fig. 7 (a). This design choice allows Mamba-3D to maintain the same computational cost while enabling deeper networks that better capture directional visual features, yielding improved performance on 3D visual tasks compared to methods that introduce complexity into the S6 internals.

Mamba-3D achieves effective processing of 3D data through specific transformations of the sequence order among different Mamba blocks, as illustrated in Fig. 7 (b) and (c). Specifically, this involves separate forward and reverse Mamba operations across three permutations of the 3D axes: "HWD", "DWH", and "DHW", as three 3D-to-1D flattening modes. Each permutation ensures contiguity along the last axis after flattening, facilitating the modeling of local and non-local dependencies across all spatial dimensions. Furthermore, the standard **Single Input** form and the proposed **Tandem (bilateral) Input** are presented respectively in the left and right parts of Fig. 7 (c).

## B  Datasets

**Further details on the datasets**  Our research utilizes three internal datasets from different institutions for training and evaluation, as well as an external dataset for independent evaluation. The data distribution and characteristics of the external dataset are prominently different from those of the internal datasets, which makes it more conducive to demonstrating the generalization capabilities of different methods. The number of samples in each dataset is shown in Tab. 4. The training and testing splits of internal datasets are designed to evenly distribute samples across different collection periods and varying lesion sizes.

The original data in each dataset consists entirely of non-contrast chest CT images with spatial resolution $D \times 512 \times 512$, where $D$ denotes the number of slices per volume, typically ranging

Table 4: **Sample counts for the datasets used in this study.**

| Dataset | Training | | | | Testing | | | |
|---|---|---|---|---|---|---|---|---|
| | Patients | Breasts | Cancerous | Non-cancerous | Patients | Breasts | Cancerous | Non-cancerous |
| Internal Inst. 1 | 328 | 656 | 341 | 315 | 80 | 160 | 82 | 78 |
| Internal Inst. 2 | 580 | 1160 | 239 | 921 | 199 | 398 | 102 | 296 |
| Internal Inst. 3 | 127 | 254 | 141 | 113 | 31 | 62 | 34 | 28 |
| External Inst. | - | - | - | - | 107 | 214 | 105 | 109 |
| Total | 1035 | 2070 | 721 | 1349 | 417 | 834 | 323 | 511 |

from several dozen to several hundred depending on acquisition settings. To ensure the reliability and accuracy of the annotations, experienced radiologists with years of expertise delineated the boundaries of breast lesions based on contrast-enhanced CT images taken concurrently for each patient. These segmentation labels were then accurately registered to the non-contrast CT (NCCT) volumes using DEEDS [25]. All malignant cases were confirmed via pathological reports, while benign or normal cases were validated either through pathological examinations or two-year clinical follow-ups.

In Fig. 8, we present the visualizations of representative data samples with lesions from each dataset, from left to right including: (a) the representative NCCT 2D slice view with a window width of 300 and window level of 50, (b) the 2D view with segmentation labels for breast and lesion, (c) the registered CECT slices to highlight the challenge of using non-contrast CT instead of contrast-enhanced CT, (d) the 3D rendered image, and (e) the 3D rendered image with segmentation labels for breast and lesion. Red annotations represents the breast, while green and blue annotations respectively represent malignant or benign lesions. The samples from the four institutions are presented consecutively from top to bottom, with two samples from each institution.

The study received approval from the Institutional Review Board (IRB) with Approval No.: B2025-235R. All procedures adhered to established ethical standards and regulations. Confidentiality of participant data was stringently maintained, with all identifying information anonymized during data collection and analysis. Regular audits were conducted to verify compliance with ethical guidelines, and any unforeseen ethical concerns that arose during the study were promptly addressed in consultation with institutional review boards.

**Pre-Stage: breast region cropping**   As described in Sec. 3.1, we use a pre-trained segmentation network to achieve coarse-grained cropping of each breast region in the Pre-Stage. Since this step does not require particularly precise segmentation results, we trained this coarse-grained segmentation model on a few hundred samples with breast masks using nnUNet [30], which achieved a high Dice score of 0.9598, demonstrating its accuracy and reliability.

To further minimize the risk of missing boundary lesions or subtle structures (*e.g.*, in obese patients or cases with blurred borders), we adopted a conservative cropping strategy: the bounding box of the predicted breast mask is expanded by 2 voxels along the D-axis and 16 voxels along the H- and W-axes. This ensures that small lesions near the periphery are retained within the cropped region. In future deployment, extremely rare failure cases of the Pre-Stage can be flagged for manual review or fallback processing.

Following this, we obtained the training sets actually used in the Main-Stage, where each sample contained two partial NCCT images corresponding to the left and right breast regions, along with the voxel-level lesion masks and breast-level cancer classification labels. For methods that do not use bilateral inputs, each side is treated as an independent sample. Additionally, we retained the breast region masks from this step as a third segmentation category, alongside the background and lesion masks, to guide effective learning in all networks, including ours and all competing methods.

## C   Implementation Details

This section elaborates on the implementation details of the Main-Stage in the overall pipeline (Fig. 2) of this study, while the the Pre-Stage has been introduced in Sec. B.

To ensure a fair comparison, all methods in this study—including those used for benchmarking and ablation studies—share identical pre-processing, post-processing, and data augmentation protocols. The only difference lies in the synchronization of randomness during data augmentation when bilateral inputs are used.

**Pre-processing**   Firstly, we unify the three-dimensional spacings of CT images through resampling, with the target values being the median spacings of all samples in all training sets, which are (3 mm, 0.748 mm, 0.748 mm). Subsequently, we adopt the normalization scheme introduced by [30] to the Hounsfield Unit (HU) values of CT images. Let $\mathbf{x}$ denote the original Hounsfield Unit (HU) value of voxels in a CT image. For all foreground voxels (*i.e.*, breast and lesion regions), we first compute the 0.5th and 99.5th percentiles of the HU distribution, denoted as $q_{0.5}$ and $q_{99.5}$, respectively. The voxel intensities are then clipped and standardized as follows:

$$\mathbf{x}_{\text{clipped}} = \min\big(\max(\mathbf{x}, q_{0.5}), q_{99.5}\big)$$
$$\mathbf{x}_{\text{norm}} = \frac{\mathbf{x}_{\text{clipped}} - \mu}{\sigma} \tag{13}$$

where $\mathbf{x}_{\text{clipped}}$ is the HU value after quantile-based clipping; $\mu$ and $\sigma$ are the mean and standard deviation of the clipped HU values within the foreground; $\mathbf{x}_{\text{norm}}$ is the final normalized HU value. This normalization procedure helps suppress noise and outliers, while ensuring a standardized input range for downstream processing.

Given that the cropped breast regions vary in size, we further crop each single breast in the Main-Stage as nnUNet [30] did, so that it has a uniform size as $56 \times 160 \times 192$ when fed into our network and all the competitors, which is slightly smaller than the median size of all single-sided samples. In cases where the input no longer contains any lesion voxels after the uniformly sized cropping, we modify the breast-level label to "*non-cancerous*" by verifying that no lesion voxels are present in the cropped region during training.

**Data augmentations**   To leverage the symmetric nature of breasts in alignment with our proposed Tandem Mamba-3D blocks, we flipped the left breast image along the X-axis (*i.e.*, the W-axis in the network's inputs and outputs) prior to applying data augmentations, thus treating it as a pseudo-right breast. During training, the cropped left and right breast images from the same patient undergo identical data augmentation procedures, which include a random combination of scaling, rotation, Gaussian noise, Gaussian blur, brightness adjustment, contrast adjustment, inversion, gamma adjustment, and three-dimensional mirroring, as proposed and planned by [30].

**Post-processing for segmentation and classification**   As commonly used in medical image segmentation [4, 30], we apply a sliding window approach during inference to handle variations in sample sizes. For segmentation, the softmax values of each voxel are restored to the original cropped breast region's size using Gaussian-weighted superposition and spacing resampling. For classification, the final score for malignancy (cancer) is taken as the highest score across all sliding windows.

In each window, mirroring is applied independently along all three dimensions, yielding eight different results. The averaged results are then used as the test-time augmentation. As mentioned earlier, the left breast is flipped to become a pseudo-right breast for bilateral inputs, and during the eight mirroring operations, the real-right and pseudo-right breasts are kept synchronized.

**Detailed configurations and hyper-parameters**   Leveraging the linear complexity of Mamba with respect to sequence length and its efficient implementation, we adopt a relatively small patch size to enhance segmentation performance. Unless otherwise specified, the default main hyper-parameters are set as follows: the patch size for the high-resolution (HR) path is $(\text{p}_{\text{d}}, \text{p}_{\text{h}}, \text{p}_{\text{w}}) = (2, 4, 4)$ with the embedding dimension of $\text{C} = 192$; the downsampling factor from HR to low-resolution (LR) path is $(\text{p}'_{\text{d}}, \text{p}'_{\text{h}}, \text{p}'_{\text{w}}) = (2, 4, 4)$ with the LR embedding dimension of $\text{C}' = 384$; the number of backbone blocks is $N = 6$; the number of decoder blocks is $M = 2$; and the masking ratio for self-supervised pre-training is $p_{\text{mask}} = 0.8$. Each Mamba layer follows the default configurations as the original Mamba [17].

In Tab. 5 and Tab. 6, we respectively summarize the training configurations and hyper-parameters used in self-supervised pre-training and downstream fine-tuning for the proposed DR-M3D and DRT-M3D networks, including their variants used in the ablation study of different patch sizes.

During inference, all methods—ours and the competing ones—produce the exact same output format in the Main-Stage, including the voxel-leval segmentation softmax scores for each $\text{D} \times \text{H} \times \text{W}$ sliding window, and window-level classification scores after applying the test-time augmentation (*i.e.*, mirroring and averaging as previously described).

Table 5: **Default configurations and hyper-parameters of DRT-M3D and its variants of different patch sizes for self-supervised pre-training.**

| Configuration & Hyper-Parameter | DR-M3D series | | | | DRT-M3D series | | | |
|---|---|---|---|---|---|---|---|---|
| | DR-M3D/16 | DR-M3D/8 | **DR-M3D** | DR-M3D/2 | DRT-M3D/16 | DRT-M3D/8 | **DRT-M3D** | DRT-M3D-S |
| optimizer | AdamW [45] with $\beta_1 = 0.9, \beta_2 = 0.999, \varepsilon = 1e{-}15$ | | | | | | | |
| weight decay | $1e{-}1$ | | | | | | | |
| learning rate schedule | cosine decay with linear warm-up | | | | | | | |
| basic learning rate | $1e{-}3$ | | | | | | | |
| minimal learning rate | $1e{-}5$ | | | | | | | |
| warm-up epochs | 5 epochs | | | | | | | |
| total epochs | 500 epochs | | | | | | | |
| total batch size | $32 = 4(\text{batch size}) \times 2(\text{ranks}) \times 4(\text{accumulation})$ | | | | $16 = 2(\text{batch size}) \times 2(\text{ranks}) \times 4(\text{accumulation})$ | | | |
| max gradient norm | 0.1 | | | | | | | |
| mixed-precision | BFloat16 | | | | | | | |
| loss function | Dual-Path MSE Loss as Eq. (4) | | | | | | | |
| Input Size ($1 \times D \times H \times W$) | $1 \times (1 \times 56 \times 160 \times 192)$ | | | | $2 \times (1 \times 56 \times 160 \times 192)$ | | | |
| Patch Size ($p_d, p_h, p_w$) | (2, 16, 16) | (2, 8, 8) | (2, 4, 4) | (2, 2, 2) | (2, 16, 16) | (2, 8, 8) | (2, 4, 4) | (2, 2, 2) |
| Low-Res Pooling Size ($p_d', p_h', p_w'$) | (2, 1, 1) | (2, 2, 2) | (2, 4, 4) | (2, 8, 8) | (2, 1, 1) | (2, 2, 2) | (2, 4, 4) | (2, 8, 8) |
| High-Res Embedding Dimension (C) | 192 | 192 | 192 | 96 | 192 | 192 | 192 | 96 |
| Low-Res Embedding Dimension (C') | 384 | | | | | | | |
| Mutual Fusion DownSampling | 3D MaxPooling with size ($p_d', p_h', p_w'$) + Linear: C $\to$ C' | | | | | | | |
| Mutual Fusion Upsampling | 3D Nearest Interpolation with size ($p_d', p_h', p_w'$) + Linear: C' $\to$ C | | | | | | | |
| Backbone Mamba-3D Blocks ($N$) | 6 | | | | | | | |
| Decoder Mamba-3D Blocks ($M$) | 2 | | | | | | | |
| Masking Chain | (2, 1, 1) | (2, 2, 2) | (2, 4, 4) | (2, 8, 8) | (2, 1, 1) | (2, 2, 2) | (2, 4, 4) | (2, 8, 8) |
| Masking Ratio ($p_{\text{mask}}$) | 0.8 | | | | | | | |

Table 6: **Default configurations and hyper-parameters of DRT-M3D and its variants of different patch sizes for downstream fine-tuning.**

| Configuration & Hyper-Parameter | DR-M3D series | | | | DRT-M3D series | | | |
|---|---|---|---|---|---|---|---|---|
| | DR-M3D/16 | DR-M3D/8 | **DR-M3D** | DR-M3D/2 | DRT-M3D/16 | DRT-M3D/8 | **DRT-M3D** | DRT-M3D-S |
| optimizer | AdamW [45] with $\beta_1 = 0.9, \beta_2 = 0.999, \varepsilon = 1e{-}15$ | | | | | | | |
| weight decay | $1e{-}1$ | | | | | | | |
| learning rate schedule | cosine decay with linear warm-up | | | | | | | |
| basic learning rate | $1e{-}3$ | | | | | | | |
| minimal learning rate | $1e{-}5$ | | | | | | | |
| warm-up epochs | 1 epoch | | | | | | | |
| total epochs | 50 epochs | | | | | | | |
| total batch size | $32 = 4(\text{batch size}) \times 2(\text{ranks}) \times 4(\text{accumulation})$ | | | | $16 = 2(\text{batch size}) \times 2(\text{ranks}) \times 4(\text{accumulation})$ | | | |
| max gradient norm | 1.0 | | | | | | | |
| mixed-precision | BFloat16 | | | | | | | |
| loss function | $\mathcal{L}_{\text{SEG}} + \lambda\mathcal{L}_{\text{CLS}}$ as Eq. (7) with $\lambda = 0.1$ | | | | | | | |
| Input Size ($1 \times D \times H \times W$) | $1 \times (1 \times 56 \times 160 \times 192)$ | | | | $2 \times (1 \times 56 \times 160 \times 192)$ | | | |
| Patch Size ($p_d, p_h, p_w$) | (2, 16, 16) | (2, 8, 8) | (2, 4, 4) | (2, 2, 2) | (2, 16, 16) | (2, 8, 8) | (2, 4, 4) | (2, 2, 2) |
| Low-Res Pooling Size ($p_d', p_h', p_w'$) | (2, 1, 1) | (2, 2, 2) | (2, 4, 4) | (2, 8, 8) | (2, 1, 1) | (2, 2, 2) | (2, 4, 4) | (2, 8, 8) |
| High-Res Embedding Dimension (C) | 192 | 192 | 192 | 96 | 192 | 192 | 192 | 96 |
| Low-Res Embedding Dimension (C') | 384 | | | | | | | |
| Mutual Fusion DownSampling | 3D MaxPooling with size ($p_d', p_h', p_w'$) + Linear: C $\to$ C' | | | | | | | |
| Mutual Fusion Upsampling | 3D Nearest Interpolation with size ($p_d', p_h', p_w'$) + Linear: C' $\to$ C | | | | | | | |
| Backbone Mamba-3D Blocks ($N$) | 6 | | | | | | | |
| Head for Segmentation | Linear: C from High-Res Path $\to$ ($3 \times p_d \times p_h \times p_w$) , "3" for (background, breast, lesion) | | | | | | | |
| Head for Classification | 3D Global AvgPooling + MLP: C' from Low-Res Path $\to$ 512 $\to$ 512 $\to$ Malignant-Logit | | | | | | | |

For the sliding window strategy, we use a maximum stride of 50% of the window size along each dimension to determine the number of required steps. If the sizes of the left and right breast regions differ, we use the larger of the two to compute the step count. For generating Gaussian importance weights in each window, we set $\sigma = 0.125$ when computing the weight tensor.

# D   Metrics

## D.1   Lesion-Level (Segmentation-based Detection) Metrics

**Dice similarity coefficient (Dice)**   We directly evaluate the voxel-level accuracy of the segmented parts using the dice similarity coefficient:

$$\text{Dice} = \frac{1}{|S|} \sum_{i=1}^{|S|} \frac{2 \times |s_{i,\text{pred}} \cap s_{i,\text{gt}}|}{|s_{i,\text{pred}}| + |s_{i,\text{gt}}|} \tag{14}$$

where $|S|$ denotes the total number of samples in the testing set; $s_{i,\text{pred}}$ is the foreground (lesion) voxel mask obtained by the model for the $i$-th sample; $s_{i,\text{gt}}$ is the corresponding ground truth. The dice similarity coefficient is inherently normalized, making it suitable for targets of varying sizes or scales, particularly for breast lesion regions in this study. Moreover, the lesion boundaries in breast CT images are often ambiguous. Consequently, employing the dice similarity coefficient that only considers the intersection and is more lenient towards minor deviations on the edges holds practical value.

**Hit-rate (H.r.)**   We define a breast-level prediction as a "hit" if the Dice score for that sample is greater than zero, otherwise as a "non-hit". A "hit" indicates that the model has successfully segmented a lesion region that overlaps with the ground truth even if only partially, which can be sufficient to prompt further clinical attention. The Hit-rate is then defined as:

$$\text{H.r.} = \frac{\left|\{s_i \in S \mid \text{Dice}(s_{i,\text{pred}}, s_{i,\text{gt}}) > 0\}\right|}{|S|} \tag{15}$$

where the numerator $\left|\{s_i \in S \mid \text{Dice}(s_{i,\text{pred}}, s_{i,\text{gt}}) > 0\}\right|$ represents the number of samples where the dice similarity coefficient is greater than zero, indicating a "hit"; the denominator $|S|$ denotes the total number of test samples.

**FROC-score (F.s.)**   For each connected region classified as the lesion class in the segmentation results, we compute its lesion probability $t$ as the mean of the softmax scores of all voxels in that region. Subsequently, we obtained the FROC curve [2, 41] based on varying probability thresholds $\tau$. The x-axis of the FROC curve represents the false positive per breast (FPPB) indicating the average number of false positive lesion regions per breast. The y-axis the FROC curve represents the detection sensitivity indicating the proportion of true lesion-containing breasts where at least one lesion is detected.

Unlike the standard ROC curve, the x-axis of the FROC curve spans $[0, \infty)$, making it unsuitable for traditional AUC computation. Therefore, based on the characteristics of the used datasets, we define the FROC-score as the average detection sensitivity at four predefined FPPBs (0.1, 0.2, 0.3, and 0.4):

$$\text{F.s.} = \frac{1}{4} \sum_{i=1}^{4} \text{LesionSensitivity}(R_{\text{pred}}, R_{\text{gt}} | \text{FPPB}_i) \tag{16}$$

where each $\text{FPPB}_i$ is associated with a specific threshold $\tau_i$ used to determine $R_{\text{pred}}$, the set of predicted lesion regions across all samples; $R_{\text{gt}}$ denotes the corresponding set of ground-truth lesion regions. Therefore, a higher FROC-score indicates better lesion detection performance with fewer false positive regions.

### D.2   Breast-Level (Classification) Metrics

**Sensitivity under fixed specificity (Sens.)**   For opportunistic breast cancer analysis, maintaining a relatively high specificity is crucial to prevent excessive false positive alerts that could unnecessarily strain medical resources. Consequently, we selected a specificity value $\alpha$ (relatively high, while also avoiding excessively extreme classification score thresholds) for each dataset to enable fair and effective comparisons of breast-level cancer classification across different methods:

$$\text{Sens.} = \left. \frac{TP}{TP + FN} \right|_{\text{Specificity}=\alpha} \tag{17}$$

Under this setting, the sensitivity values may vary across methods, reflecting their ability to detect true positive cases at the same high specificity level. A higher sensitivity suggests more effective identification of true positives without increasing the false alarm rate.

**Area Under the receiver operating characteristic Curve (AUC)**   Apart from the sensitivity under high specificity, we also evaluate the overall classification performance using the AUC metric, defined as:

$$\text{AUC} = \int_0^1 \text{Sensitivity}(\beta) \, d\beta \tag{18}$$

where $\beta = 1 - \text{Specificity}$ represents the false positive rate of breast-level cancer classification ranging from 0 to 1, which differs from the definition of FPPB used in lesion-level metrics.

## E   Additional Experiments

**More hyper-parameter tuning**   In addition to the ablation studies in the main paper, we also conducted experiments on several other key hyper-parameters, including the patch size in the D (slice of CT scans) dimension ($p_d$), the embedding dimensions (C, C$'$), and the depth of the backbone ($N$).

As shown in Tab. 7, a smaller patch size in the D dimension, which better aligns with the larger spacing (3 mm vs. 0.748 mm in H and W dimensions), yields improved segmentation performance. However, this also leads to an increase in computational cost. To keep the computational cost (FLOPs) similar to the strong baseline (nnUNet [30]), $p_d$ = 2 is used as the default. For the ablation study on patch size in the H and W dimensions, please refer to the main paper Sec. 4.3.

As demonstrated in Tab. 8, increasing the High-Resolution Path's embedding dimension C significantly improves the voxel-level metric (Dice), while increasing the Low-Resolution Path's embedding dimension C$'$ can markedly enhance the breast-level metrics (Sens. and AUC). To balance parameter count and computational overhead, a moderate configuration (C = 192 and C$'$ = 384) is adopted.

According to Tab. 9, Dice and FROC-score are relatively sensitive to the depth of the backbone ($N$), with significant improvements observed as the network deepens. In contrast, the classification part reaches a near-optimal trade-off between complexity and performance when $N$ = 6. At this setting, the model's parameter count (45.71M) and computational load (1162.65G FLOPs) are comparable to those of the strong baseline nnUNet [30], which has 31.00M parameters and 1185.24G FLOPs.

**Comparison of other 1D sequence construction strategies with tandem input**   As described in the main text, the tandem input strategy first places the left and right 3D regions into the pseudo-3D volume and then flattens them into a 1D token sequence. This approach is actually equivalent to independently flattening the tokens of the left and right breasts along the prioritized axis and then concatenating them to form the input sequence. To further validate the superiority of this strategy, we compare it with three alternative variants: 1. last-axis-concat: Concatenate along the last (prioritized) axis (*i.e.*, D-, H-, W-axis for HWD, DWH, DHW scanning, respectively) before flattening. 2. W-axis-concat: Always concatenate along the W-axis before flattening. 3. interleaving: Interleave tokens from right and left breasts (*i.e.*, R1, L1, R2, L2, ... ; Rx/Lx denotes the x-th token in the flattened right/left sequence). Results are shown in Tab. 10.

Notably, our tandem input design ensures that all tokens from one side are processed before those from the other within the Mamba layer, and also maintains an approximately constant distance between anatomically symmetric locations, regardless of scanning order. This facilitates effective learning and comparison of bilateral features. In contrast, all alternative variants introduce some degree of mixing between left and right tokens, which leads to information confusion—most notably in the Interleaving variant, where the alternation of tokens from both sides results in the greatest disruption.

**Bilateral input on CNN-based UNet**   In Sec. 4.3 of the main paper, we have attempted to apply the bilateral input concept to Vision Transformers [14] and verified the effectiveness of this idea, yet DRT-M3D achieves substantially better results. Here, we further attempt to apply the bilateral input form to CNN-based UNet-like networks, as also shown in Tab. 10. The "UNet-W-axis-concat" method concats bilateral breasts along the W axis as in the original NCCT images; the "UNet-channel-concat" method uses bilateral breast inputs as two channels, with the left breast flipped as a pseudo-right breast; the "UNet-CLS-head-merge" method processes unilateral breasts in the UNet, with features for classification concatenated, thereby mixing the bilateral information in the classification part.

CNNs are generally incapable of modeling long-range features, as their local receptive fields require stacking many convolutional layers to capture larger spatial extents, making them less effective than our proposed DRT-M3D. Thus, simply concatenating left and right views along the W-axis ("UNet-W-axis-concat") is not effective for explicit bilateral modeling. Although stacking these views as channels ("UNet-channel-concat") may seem like a reasonable alternative, breast symmetry is not strictly voxel-to-voxel—due to natural anatomical differences, positional variation, and imaging variability—so this approach can mix features from non-corresponding regions and ultimately introduce negative effects, leading to the worst performance. Moreover, simply concatenating features from the two views at the network head ("UNet-CLS-head-merge") does not allow the model to effectively exploit their spatial correspondence.

**Comparison of model efficiency**     Tab. 12 provides the GPU memory usage, training time, and inference time for all major competing methods. The reported memory usage is measured per training sample, corresponding to one breast region. For bilateral methods, although each training sample contains two breast regions by design, we set the batch size to half that of their unilateral counterparts; this ensures that memory usage per breast region is fairly comparable across all methods. Training time reflects the sum of both pre-training and downstream fine-tuning phases (using two NVIDIA A100 80GB GPUs). Inference time is measured per patient (using a single NVIDIA A100 80GB GPU).

In addition to comparing the performance and computational cost of each method using GLOPs in the main paper, we assess the practical deployment efficiency by replacing FLOPs with inference time for each patient. The results are shown in Fig. 9.

**Additional visualization results for qualitative comparison**     In Fig. 10,we present additional qualitative comparisons between the proposed DRT-M3D and competing methods, which serve as a supplement to the results shown in Fig. 4 of the main paper.

**Delta visualization for the Tandem Input mechanism**     To better illustrate the advantages introduced by the Tandem (bilateral) Input mechanism in DRT-M3D, we visualize the Delta values ($\mathbf{\Delta}$, as described in Sec. A.1) of the Mamba-3D blocks in both DRT-M3D and DR-M3D. This allows a direct comparison of their responsiveness to different regions in the chest NCCT images, particularly along the low-resolution (LR) path. A larger step $\mathbf{\Delta}_t$ indicates a faster change in the hidden state $\mathbf{h}_t$, implying a greater influence of the current token $\mathbf{x}_t$ on the overall system dynamics. In this sense, $\mathbf{\Delta}_t$ can be viewed as a soft analog to attention weights in Transformers [59].

To better observe responses across all layers, we use a simplified setup with $N = 1$ for both models, meaning each high-resolution (HR) and low-resolution (LR) path contains a single Mamba-3D block composed of six internal Mamba layers (as detailed in Sec. A.2). We generate the Delta visualization by averaging first over all channels and then across the six layers.

The resulting visualizations are presented in Fig. 11. The Delta responses on the HR path mainly reflect the model's focus on local textures and anatomical details, which aligns with its primary role for the segmentation task. Meanwhile, the LR path is designed to focus on capturing bilateral context with shorter sequence length when Tandem Input mechanism is used (*i.e.*, in DRT-M3D), so the Delta visualization of the LR path shows clear cross-breast clues. This bilateral awareness enhances cancer classification reliability. Notably, the model's ability to focus on corresponding regions across both breasts is learned rather than derived from explicit coordinate symmetry, which is often lacking due to anatomical variation.

**Results on contrast-enhanced CT data**     Here, we conduct additional experiments on contrast-enhanced CT (CECT) as the approximate upper bound of performance using NCCT data for this task. The data sources are as described in Sec. B. Since the corresponding CECT has been registered with NCCT, the segmentation labels used by both are exactly the same, with only the images being different. The intuitive differences between NCCT and CECT in breast lesions are shown in Fig. 8. The experimental results on internal and external datasets are presented in Tab. 11.

## F   Discussion of Broader Impacts

As discussed in the main contributions of this paper, the proposed approach aims to enhance opportunistic breast cancer analysis techniques while minimizing economic costs and radiation risks encountered by patients. This advancement holds the potential to provide earlier diagnosis and treatment options to a larger population of potential breast cancer patients, thereby significantly impacting their health outcomes.

Regarding potential negative impacts, it is important to note that this proposed method has not yet been deployed in practical healthcare systems. Nonetheless, it is foreseeable that unavoidable false positive cases may impose an additional burden on healthcare systems. Despite possible concerns, the anticipated benefits of early detection, classification, further examination, and subsequent treatment are expected to outweigh these drawbacks, resulting in an overall positive impact.

Table 7: **Ablation study on the patch size in the** $D$ **(slice of CT scans) dimension.** All the experiments were conducted based on the default DRT-M3D model, with modifications applied to $p_d$. The gray row represents the default settings. (H.r.: Hit-Rate, F.s.: FROC-Score, Spec.: Specificity, Sens.: Sensitivity)

| Variants | Complexity | | Internal (Spec. = 0.9627) | | | | | External (Spec. = 0.9083) | | | | |
|---|---|---|---|---|---|---|---|---|---|---|---|---|
| $p_d, p_h, p_w$ | Params (M) | FLOPs (G) | Dice | H.r. | F.s. | Sens. | AUC | Dice | H.r. | F.s. | Sens. | AUC |
| nnUNet [30] | 31.00 | 1185.24 | 0.6259 | 0.9127 | 0.8548 | 0.7844 | 0.9536 | 0.5668 | 0.8889 | 0.8173 | 0.6286 | 0.8889 |
| 4, 4, 4 | 45.74 | 582.65 | 0.6517 | 0.9214 | 0.9065 | 0.8242 | 0.9737 | 0.5867 | 0.8975 | 0.8652 | 0.8309 | 0.9347 |
| 2, 4, 4 | 45.71 | 1162.65 | 0.6590 | 0.9229 | 0.9145 | 0.8578 | 0.9768 | 0.5948 | 0.9117 | 0.8766 | 0.8762 | 0.9371 |
| 1, 4, 4 | 45.70 | 2322.66 | 0.6651 | 0.9243 | 0.9159 | 0.8654 | 0.9729 | 0.6143 | 0.9211 | 0.9142 | 0.9002 | 0.9439 |

Table 8: **Ablation study on the embedding dimensions of two resolution paths.** All the experiments were conducted based on the default DRT-M3D model, with modifications applied to $C$ for the High-Resolution Path and $C'$ for the Low-Resolution Path. The gray row represents the default settings. (H.r.: Hit-Rate, F.s.: FROC-Score, Spec.: Specificity, Sens.: Sensitivity)

| Variants | | Complexity | | Internal (Spec. = 0.9627) | | | | | External (Spec. = 0.9083) | | | | |
|---|---|---|---|---|---|---|---|---|---|---|---|---|---|
| $C$ | $C'$ | Params (M) | FLOPs (G) | Dice | H.r. | F.s. | Sens. | AUC | Dice | H.r. | F.s. | Sens. | AUC |
| nnUNet [30] | | 31.00 | 1185.24 | 0.6259 | 0.9127 | 0.8548 | 0.7844 | 0.9536 | 0.5668 | 0.8889 | 0.8173 | 0.6286 | 0.8889 |
| 96 | 192 | 12.51 | 339.74 | 0.6404 | 0.9214 | 0.8767 | 0.8073 | 0.9674 | 0.5842 | 0.9093 | 0.8644 | 0.8202 | 0.9215 |
| 192 | 192 | 19.37 | 1075.07 | 0.6584 | 0.9083 | 0.9028 | 0.8073 | 0.9683 | 0.5920 | 0.9093 | 0.8653 | 0.8282 | 0.9319 |
| 192 | 384 | 45.71 | 1162.65 | 0.6590 | 0.9229 | 0.9145 | 0.8578 | 0.9768 | 0.5948 | 0.9117 | 0.8766 | 0.8762 | 0.9371 |
| 192 | 768 | 148.15 | 1498.35 | 0.6636 | 0.9258 | 0.9094 | 0.8624 | 0.9757 | 0.6015 | 0.9164 | 0.8795 | 0.8789 | 0.9414 |
| 384 | 768 | 175.76 | 4258.06 | 0.6716 | 0.9170 | 0.9127 | 0.8670 | 0.9771 | 0.6036 | 0.9235 | 0.8955 | 0.8762 | 0.9374 |

Table 9: **Ablation study on the number of backbone blocks.** All the experiments were conducted based on the default DRT-M3D model, with modifications applied to $N$. The gray row represents the default settings. (H.r.: Hit-Rate, F.s.: FROC-Score, Spec.: Specificity, Sens.: Sensitivity)

| Variants | Complexity | | Internal (Spec. = 0.9627) | | | | | External (Spec. = 0.9083) | | | | |
|---|---|---|---|---|---|---|---|---|---|---|---|---|
| $N$ | Params (M) | FLOPs (G) | Dice | H.r. | F.s. | Sens. | AUC | Dice | H.r. | F.s. | Sens. | AUC |
| nnUNet [30] | 31.00 | 1185.24 | 0.6259 | 0.9127 | 0.8548 | 0.7844 | 0.9536 | 0.5668 | 0.8889 | 0.8173 | 0.6286 | 0.8889 |
| 1 | 8.47 | 196.28 | 0.6382 | 0.9214 | 0.8963 | 0.8165 | 0.9677 | 0.5769 | 0.9069 | 0.8674 | 0.8415 | 0.9252 |
| 3 | 23.37 | 582.83 | 0.6483 | 0.9214 | 0.8996 | 0.8318 | 0.9718 | 0.5925 | 0.9093 | 0.8760 | 0.8602 | 0.9331 |
| 6 | 45.71 | 1162.65 | 0.6590 | 0.9229 | 0.9145 | 0.8578 | 0.9768 | 0.5948 | 0.9117 | 0.8766 | 0.8762 | 0.9371 |
| 9 | 68.06 | 1742.48 | 0.6615 | 0.9228 | 0.9126 | 0.8486 | 0.9744 | 0.5960 | 0.9022 | 0.8809 | 0.8621 | 0.9357 |
| 12 | 90.40 | 2322.30 | 0.6659 | 0.9243 | 0.9120 | 0.8440 | 0.9738 | 0.5974 | 0.9046 | 0.8866 | 0.8629 | 0.9373 |

Table 10: **Evaluation of several DRT-M3D variants for Tandem Input effectiveness, together with results of attempts using the bilateral input form on the CNN-based UNet.** Representative unilateral (nnUNet, DR-M3D) models are also provided for comparison. (H.r.: Hit-Rate, F.s.: FROC-Score, Spec.: Specificity, Sens.: Sensitivity)

| Method | Internal (Spec. = 0.9627) | | | | | External (Spec. = 0.9083) | | | | |
|---|---|---|---|---|---|---|---|---|---|---|
| | Dice | H.r. | F.s. | Sens. | AUC | Dice | H.r. | F.s. | Sens. | AUC |
| nnUNet [30] | 0.6259 | 0.9127 | 0.8548 | 0.7844 | 0.9536 | 0.5668 | 0.8889 | 0.8173 | 0.6286 | 0.8889 |
| **DR-M3D (ours)** | 0.6564 | 0.9228 | 0.9027 | 0.8073 | 0.9690 | 0.5909 | 0.8974 | 0.8632 | 0.7207 | 0.9082 |
| UNet-W-axis-concat | 0.6225 | 0.8940 | 0.8597 | 0.7431 | 0.9469 | 0.5696 | 0.8974 | 0.8120 | 0.6190 | 0.8936 |
| UNet-channel-concat | 0.5888 | 0.8958 | 0.8400 | 0.7798 | 0.9498 | 0.5220 | 0.8461 | 0.7760 | 0.5143 | 0.8839 |
| UNet-CLS-head-merge | 0.6208 | 0.9114 | 0.8575 | 0.8211 | 0.9559 | 0.5643 | 0.9059 | 0.8057 | 0.7333 | 0.9053 |
| DRT-M3D-last-axis-concat | 0.6536 | 0.9127 | 0.9083 | 0.8440 | 0.9693 | 0.5808 | 0.9060 | 0.8729 | 0.8190 | 0.9206 |
| DRT-M3D-W-axis-concat | 0.6551 | 0.9039 | 0.9062 | 0.8532 | 0.9732 | 0.5923 | 0.9060 | 0.8708 | 0.8476 | 0.9258 |
| DRT-M3D-interleaving | 0.6414 | 0.9083 | 0.9018 | 0.8211 | 0.9650 | 0.5749 | 0.8974 | 0.8644 | 0.7810 | 0.9162 |
| **DRT-M3D (ours)** | **0.6590** | **0.9229** | **0.9145** | **0.8578** | **0.9768** | **0.5948** | **0.9117** | **0.8766** | **0.8762** | **0.9371** |

Table 11: **Results on the corresponding contrast-enhanced CT (CECT) scans as the approximate upper bound of performance using NCCT scans for this task..** (H.r.: Hit-Rate, F.s.: FROC-Score, Spec.: Specificity, Sens.: Sensitivity)

| Method | Internal (Spec. = 0.9627) | | | | | External (Spec. = 0.9083) | | | | |
|---|---|---|---|---|---|---|---|---|---|---|
| | Dice | H.r. | F.s. | Sens. | AUC | Dice | H.r. | F.s. | Sens. | AUC |
| nnUNet [30] | 0.7078 | 0.9563 | 0.9323 | 0.8257 | 0.9724 | 0.6860 | 0.9487 | 0.9068 | 0.9238 | 0.9288 |
| **DR-M3D (ours)** | 0.7252 | 0.9476 | 0.9436 | 0.9129 | 0.9736 | 0.6878 | 0.9345 | 0.9146 | 0.9302 | 0.9376 |
| DRT-M3D (ours) | 0.7263 | 0.9534 | 0.9479 | 0.9480 | 0.9795 | 0.7018 | 0.9544 | 0.9337 | 0.9365 | 0.9497 |

Table 12: Training memory usage, training time, and inference time for all major competing methods. Memory usage is measured per breast region. Training time sums pre-training and downstream fine-tuning (2×A100 GPUs), and inference time is per patient (1×A100 GPU).

| Method | Training Memory (GB) | Training Time (h) | Inference Time (s) |
|---|---|---|---|
| nnUNet [30] | 3.826 | 31.0 | 0.126 |
| VNet [48] | 3.489 | 28.9 | 0.112 |
| swinUNETR [24] | 11.975 | 36.8 | 0.139 |
| nnFormer [69] | 2.063 | 10.9 | 0.029 |
| 3D UX-Net [34] | 8.754 | 67.0 | 0.251 |
| MedNeXt [52] | 12.095 | 43.1 | 0.194 |
| U-Mamba-Bot [46] | 6.105 | 44.4 | 0.196 |
| U-Mamba-Enc [46] | 11.810 | 77.9 | 0.291 |
| EM-Net [6] | 6.218 | 34.9 | 0.116 |
| SegMamba [64] | 9.580 | 66.2 | 0.318 |
| ViT-S/8 [14] | 4.611 | 22.1 | 0.034 |
| T-ViT-S/8 | 4.629 | 26.9 | 0.047 |
| **DR-M3D/8 (ours)** | 2.565 | 14.6 | 0.049 |
| **DRT-M3D/8 (ours)** | 2.581 | 15.7 | 0.059 |
| **DR-M3D (ours)** | 9.187 | 30.4 | 0.143 |
| **DRT-M3D (ours)** | 9.202 | 34.2 | 0.171 |

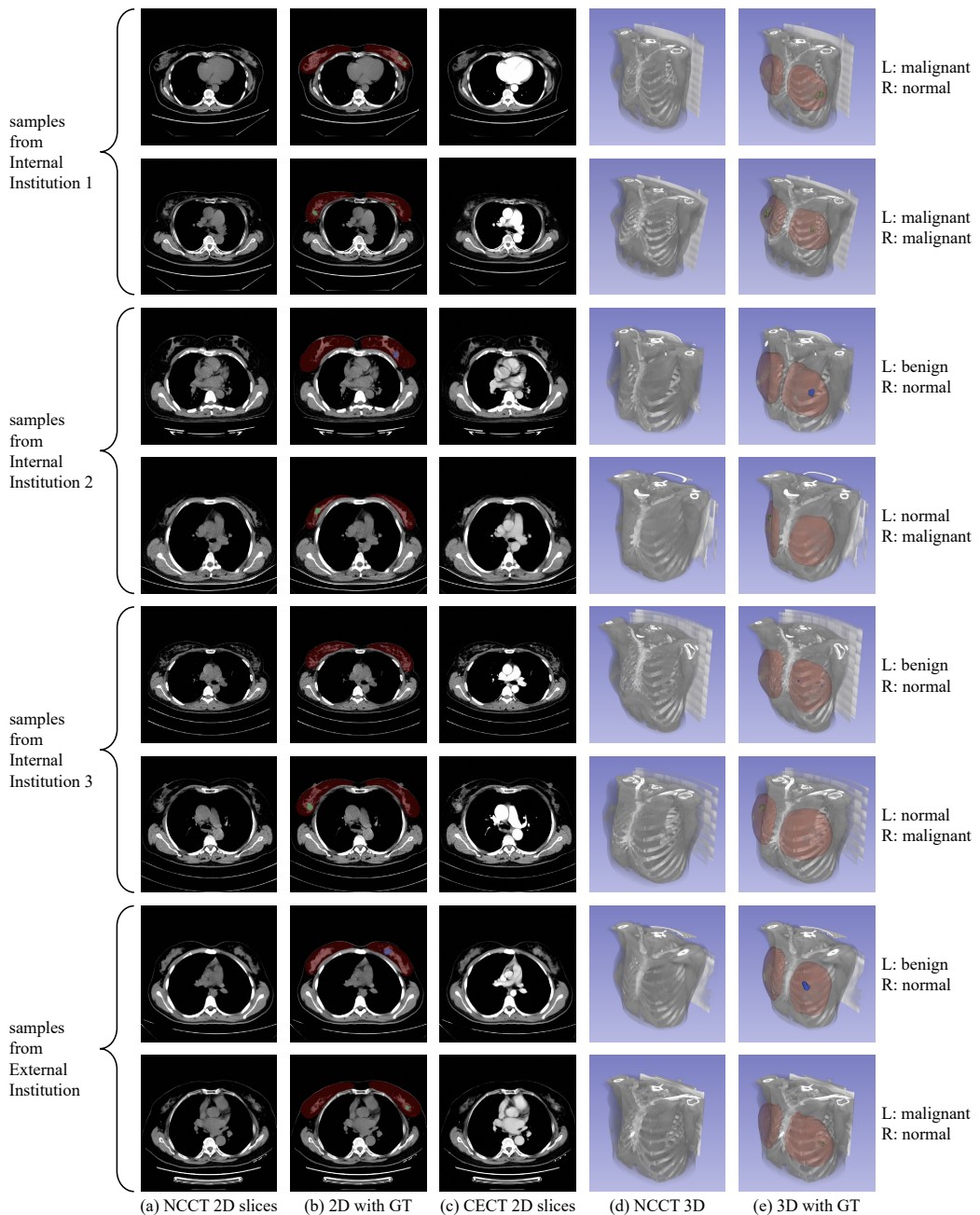

Figure 8: **Visualization of representative data samples from the four datasets used in this study.** (a), (b) and (c) respectively show typical NCCT 2D slices, their corresponding segmentation ground truth, and the corresponding CECT slices that have already been registered to NCCT. (d) and (e) present 3D renderings of the NCCT images and their segmentation ground truth. Green and blue indicate malignant and benign lesions, respectively. The breast-level classification labels are shown on the right side of each row. Note that the 2D slices are viewed in the foot-to-head direction, meaning the left side of each image corresponds to the right side of the human body.

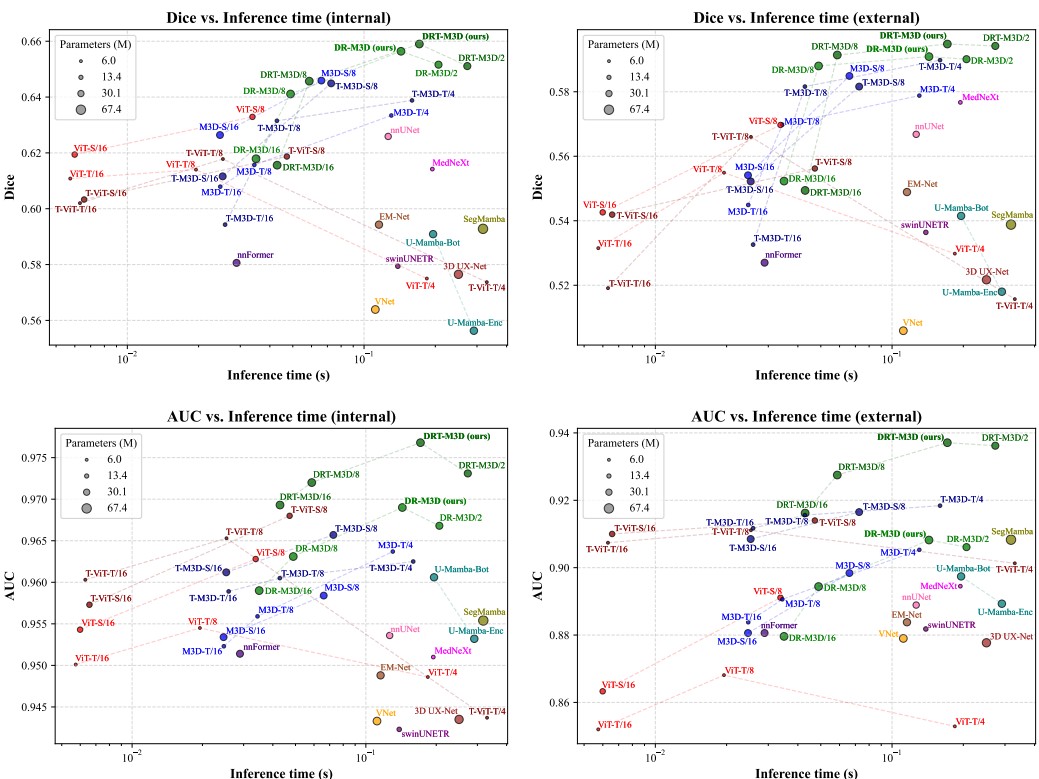

Figure 9: **Comparison of Dice and AUC vs. Inference time for internal and external datasets.** Each inference time is measured based on one bilateral (or two unilateral) input. ViT: vanilla Vision Transformer, M3D: vanilla Mamba-3D, T-~: tandem (bilateral) input form, ~/p: patch size of (2, p, p).

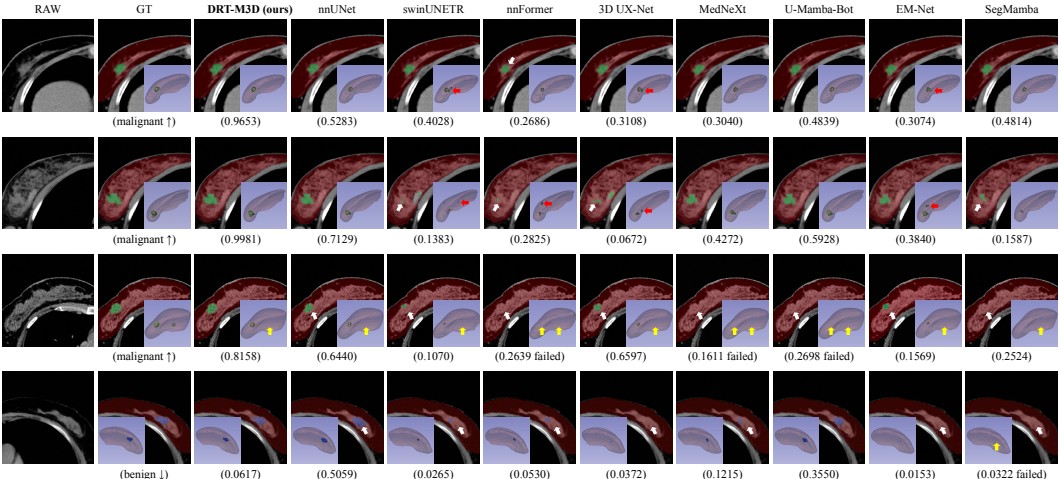

Figure 10: **More visualizations as a supplement to Fig. 4 in the main paper.** Red, green, and blue masks represent breast regions, malignant, and benign lesions, respectively. White, yellow, and red arrows mark pronounced segmentation errors in the slice, missed lesions, and segmentation of non-existent lesions, respectively. Predicted malignancy scores are shown for each case (higher for malignant, lower for benign). "Failed" denotes cases where none of the true lesions were localized (*i.e.*, a non-hit under the Hit-Rate metric).

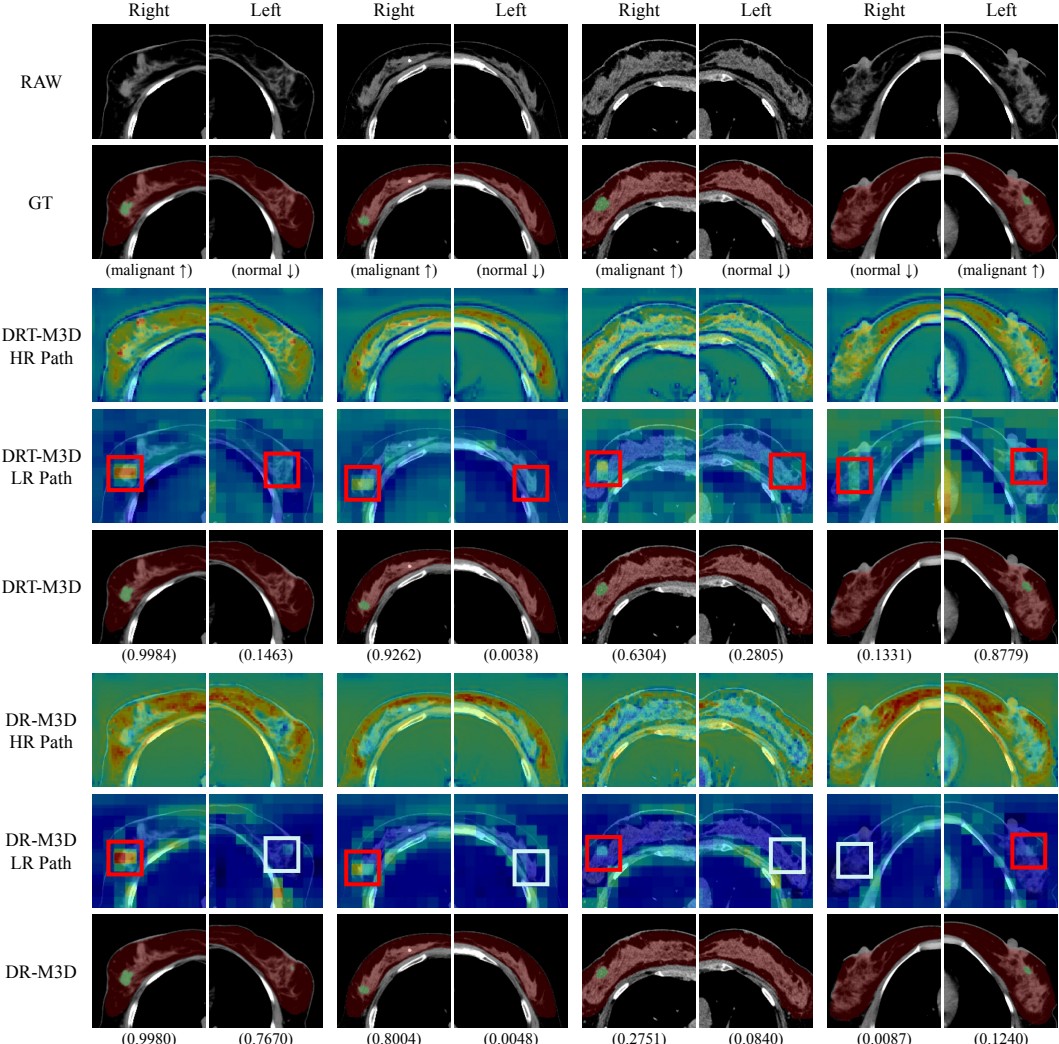

Figure 11: **Delta visualization of Mamba-3D blocks in DRT-M3D and DR-M3D.** Each column shows a test sample pair from three internal datasets and one external dataset; left and right images correspond to the right and left breast, respectively. Rows display (1) raw CT, (2) ground truth (red: breast, green: lesion, below: classification label), (3–4) Delta heatmaps of DRT-M3D (HR and LR paths), (5) DRT-M3D predictions, (6–7) Delta heatmaps of DR-M3D (HR and LR paths), and (8) DR-M3D predictions. The focus is on LR heatmaps: DRT-M3D, with the Tandem Input mechanism, highlights lesion areas and contralateral counterparts (red boxes), leading to superior classification results. In contrast, DR-M3D fundamentally lacks this ability to leverage contralateral information (light cyan boxes).

