# OpenReview forum: "Dual-Res Tandem Mamba-3D: Bilateral Breast Lesion Detection and Classification on Non-contrast Chest CT"
_NeurIPS.cc/2025/Conference — NeurIPS 2025 poster_

### Official Review · Reviewer_9b6M · 2025-06-29

**Clarity:** 3
**Significance:** 3
**Originality:** 3
**Rating:** 4
**Confidence:** 3

**Summary:**

This paper introduces DRT-M3D, a multitask, dual-resolution network utilizing Mamba-3D blocks to perform both segmentation-based lesion detection and malignancy classification on non-contrast chest CT (NCCT) scans. The authors provide comprehensive results for a methodology that can be considered a great contribution. However, the text is complex and should be simplified for all readers.

**Questions:**

I mentioned my comments in the weakness section. Please address those comments.

**Ethical Concerns:**

["NO or VERY MINOR ethics concerns only"]

**Final Justification:**

Thank you for the responses. Based on your feedback, I have increased the originality rating to 'good.' I am maintaining my overall rating as I believe the paper has potential for acceptance.

**Limitations:**

NCCT is not optimized for breast imaging (unlike mammography or DCE-MRI), and lesions may be less visible or ambiguous without contrast.

**Paper Formatting Concerns:**

I think it is okay.

**Quality:**

3

**Strengths And Weaknesses:**

Strengths:
- This manuscript explores NCCT, which is underutilized for breast cancer detection. This is a novel and clinically relevant use case.
- I think this methodology with comprehensive experiments presented in this paper can be considered novel.
- This manuscript outperforms several SOTA methods across multiple internal and external datasets.

Weaknesses:
- This manuscript is complex to follow. Some sections are too math-heavy without sufficient intuition.
- Figures should also be simplified. I suggest adding a figure to introduce the framework in a simple way.
- The text should be double-checked to fix some mistakes. For example, Figure 1 is not referenced or explained in the text. All figures should be explained in detail.

---

> ### Author Rebuttal · Authors · 2025-07-30
>
> **W1**: concerns about math-heavy sections without sufficient intuition
>
> **A**: We appreciate the comment regarding the need for more intuition in math-heavy sections.
>
> Generally speaking, our method aligns with the way radiologists interpret breast images: they analyze fine local details to identify subtle lesions, while also considering global appearance and bilateral symmetry to detect abnormalities.
>
> Regarding detailed designs, the intuition can be summarized as:
>
> 1. *Zoom-in & Zoom-out (Dual-Resolution Paths):*
>    - **Intuition**: Radiologists “zoom in” to inspect suspicious regions in detail for subtle lesions, and “zoom out” to evaluate the overall breast structure and context.
>    - **Our model**: We use two parallel paths: a high-resolution “zoom-in” path (small patches) for fine-grained segmentation, and a low-resolution “zoom-out” path (large patches) for broader contextual classification. These paths exchange information via the Mutual Fusion mechanism (see Figure 3(a)), emulating how radiologists integrate local and global cues in clinical reading.
> 2. *Cross-breast Comparison (Bilateral/Tandem Input):*
>    - **Intuition**: Radiologists routinely compare the left and right breasts, since subtle asymmetry is often a key sign of disease.
>    - **Our model**: Our “Tandem Input” approach processes both breasts together, enabling the model to directly learn cross-breast differences. The input sequence aligns anatomically symmetric locations between sides (see Figure 3(b)), mirroring how radiologists visually match corresponding regions.
> 3. *Multi-orientation Assessment (3D Patch modeling using Mamba-3D):*
>    - **Intuition**: Radiologists interpret scans from multiple anatomical planes to understand tumors’ shape and context in 3D.
>    - **Our model**: Mamba-3D prioritizes scanning along different axes, allowing structural information from multiple directions to be integrated, thus mimicking the multi-orientation analysis and comprehensive assessment performed in real clinical evaluation.
>
> **W2**: concerns regarding figure complexity
>
> **A**: Thank you for your suggestion regarding figure clarity. We agree that a simplified, high-level schematic would make our framework and overall workflow easier to understand. In the camera-ready version, we will revise Figure 2 to provide a more streamlined summary of the complete framework, and will further simplify other figures as needed to enhance clarity.
>
> **W3**: omission of Figure 1 citation in the main text
>
> **A**: We apologize for the oversight. Figure 1 should have been referenced in the first paragraph of the Introduction section, and we will correct the text to ensure it is properly cited and explained in the revised version. We will also systematically review the manuscript to check for other potential omissions or unclear figure references.
>
> All figures are explained in detail in their captions and the corresponding sections of the main text; however, we are happy to further clarify any figure or section that remains ambiguous. We also welcome any specific questions or requests for clarification during the response phase.
>
> **L1**: concerns about NCCT not optimized for breast imaging
>
> **A**: We fully recognize that NCCT is not optimized for breast imaging, and that lesion visibility is inherently limited compared to dedicated modalities or contrast-enhanced CTs. This makes our task particularly challenging and, to our knowledge, relatively under-explored in the literature.
>
> However, we believe this is precisely where the value of our work lies: by leveraging chest CT scans that originally conducted for lung or heart screening, our approach enables opportunistic breast cancer detection without additional radiation exposure or economic burden.
>
> Contrast-enhanced CT can have slightly better performance on our task but it needs contrast injection and generally not common, not appropriate for screening. To further enrich our evaluation, we have included additional experiments on contrast-enhanced CT (see our response to ***Reviewer oeU9’s W4***), where performance is stronger as expected. Nevertheless, our model achieves robust and clinically meaningful results on NCCT scans alone, as demonstrated across multi-institutional cohorts.
>
> Mammography and ultrasound can be seen as a specific examination tool for breasts, not aligned with our opportunistic screening intention. Dynamic contrast-enhanced MRI (DCE-MRI) can be deemed as a diagnostic tool, also not suitalble for oppotunistic screening.
>
> Moreover, we have also conducted a supplemental reader study on newly collected NCCT samples from anthor region (see our response to ***Reviewer THvZ’s Q1***), further supporting the practical value of our method in real-world clinical settings.

---

> > ### Comment · Area_Chair_cCKV · 2025-08-02
> > **Rebuttal**
> >
> > Dear Reviewer 9b6M
> >
> > Thanks for your review. Could you read the authors' rebuttal and update your review if needed?
> >
> > Best regards
> > Your AC

---

> ### Author Response · Authors · 2025-08-06
>
> Thank you for your time and effort in reviewing our paper. As there have been no further comments during the discussion period so far, we would like to check if there are any remaining questions or concerns we can address. Please feel free to let us know if any additional clarification is needed.

---

> > ### Comment · Area_Chair_cCKV · 2025-08-06
> > **Last chance to discuss the paper with the authors**
> >
> > Dear Reviewer 9b6M
> >
> > As we are approaching the end of the discussion phase, it'd be great if you could have a look at the authors' rebuttal, interact with the authors and then reach a final decision.
> >
> > Best regards,
> > Your AC.

---

> ### Author Response · Authors · 2025-08-08
> **Follow-up on Discussion and Acknowledgement**
>
> Thank you for your efforts in reviewing our submission and for taking the time to read our rebuttal.
>
> We have noticed, based on the last update time in the system, that a final justification has been provided. However, to fulfill the conference’s discussion requirements, we would greatly appreciate it if you could briefly comment on whether our rebuttal has addressed your concerns, and then complete the mandatory acknowledgement in the reviewing system.
>
> If you have any further questions or concerns, please feel free to let us know. We will do our best to respond before the deadline.
>
> Thank you again for your time and support.

---

### Official Review · Reviewer_y5XU · 2025-07-02

**Clarity:** 4
**Significance:** 3
**Originality:** 3
**Rating:** 5
**Confidence:** 4

**Summary:**

This paper proposed Dual-Res Tandem Mamba-3D (DRT-M3D), a 3D mamba model for breast cancer detection from non-contrast CT data. It take the efficiency advantage of the Mamba model to process very long 3D voxel sequence and make use of the multi-view relationship of the breast to improve the detection. The proposed method is evaluated on 4 datasets and outperforms existing baselines with a notable gap according to the detailed evaluation.

**Questions:**

The reviewer wonders how is the training speed of the proposed method comparing with other baselines, will it have a much longer training time comparing with other methods?

**Ethical Concerns:**

["NO or VERY MINOR ethics concerns only"]

**Final Justification:**

After reading the rebuttal and comments from other review, my concerns have been addressed an I would suggest acceptance for this paper.

**Limitations:**

Limitation was discussed in the paper.

**Paper Formatting Concerns:**

The font size in most of the Figure is too small, especially Figure 1 and all the line plot.

**Quality:**

3

**Strengths And Weaknesses:**

Strength:

1. The proposed method was designed for a task where high resolution 3D voxel segmentation is needed, which properly validated the reason of using Mamba model. Different regular 2D cases, the proposed method is often operating on sequence over 10k tokens (according to the paper). The experimental results also confirmed the computational efficiency of the proposed model under the given settings.
2. The proposed method seems to be the very first attempt to make use the multi-view relationship of the breast for the sick of breast cancer detection in NCCT.
3. The experimental evaluation is very convincing and the paper is well-written and overall easy to follow. The evaluation has demonstrates the advantage of the proposed method under various settings, and the proposed method seems to be the best in general.

Weakness:

1. The proposed method has a much better computational efficiency (FLOPs) comparing with regular transformer-based method. However, its computational speed seems to be much slower than the baselines according to the Figure 9 in the supplement even during the inference.
2. In section 3.3, the input from 2 views are placed as a pseudo-3D volume and then flattened before feeding to the mamba module. However, the reviewer didn’t see the reason of doing this. When the multi-view input was arranged as it is in Figure 3, the output sequence should be the same as simply concatenate them separately, regardless to the traverse order in 3D space. Since all the padded tokens are skipped, and there is no positional embedding, it seems to be an unnecessary operation to the reviewer. Some more clarification will help better understand this.
3. While applying multi-view information in 3D NCCT breast cancer detection is a new setting, the idea of feeding multi-view inputs to an attention-style model to obtain a cross-view relationship is not that novel. For instance, [A] also feed the multi-view breast mammography to the transformer model with cross-attention to learn cross-view information. Still, this is a minor weakness, and the reviewer is still very glad to see this work.
4. Another small issue is about the fairness of the evaluation of the main results. The proposed method was first pre-trained with mask reconstruction task for a long time and then fine-tuned for detection task. However, it seems that the baselines (except ViT baseline) are not pre-trained like this. Figure 5 suggests that pre-training is very important for the final results. So, it is unclear if the performance improvement is mainly from the pre-training, or better design.

[Reference]

- [A] Sun, Zizhao, et al. "Transformer based multi-view network for mammographic image classification." *International conference on medical image computing and computer-assisted intervention*. Cham: Springer Nature Switzerland, 2022.

---

> ### Author Rebuttal · Authors · 2025-07-30
>
> **W1**: concerns regarding the slower computing speed compared to better FLOPs
>
> **A**: Thank you for raising this point. We note that transformer-based models benefit from highly optimized implementations in current deep learning libraries (e.g., Pytorch 2.0+), which can result in surprisingly fast runtimes despite high FLOPs.
>
> While our method does not yet benefit from the same kernel-level optimization available for mainstream Transformer models, it achieves practical inference speeds (~0.1s per CT scan, as shown in Figure 9), making it suitable for near real-time application in clinical practice.
>
> We have also provided a detailed table of model efficiency in our response to ***Q1***.
>
> **W2**: some more clarification of creating the pseudo-3D volume and then flattening it
>
> **A**: We are happy to provide detailed clarification for the operation illustrated in Figure 3(b). The actual construction of the 1D sequence from the pseudo-3D volume in our design is indeed consistent with the reviewer's understanding, i.e., it is equivalent to independently flattening the left and right breasts' tokens along the prioritized axis and then concatenating them to form the input sequence.
>
> We chose to represent this process as a pseudo-3D volume (as shown in Figure 3(b)) purely for intuitive visualization and explanation, so as to avoid any potential misunderstanding that the left and right sides are first concatenated along a specific axis before flattening, which will lead to interleaving of tokens from the left and right sides under prioritized scanning along specific axes. To clarify this, we performed additional comparative experiments, as described below.
>
> **Variant-1: concatenate along the prioritized axis (i.e., D-, H-, W-axes for HWD, DWH, DHW scanning, respectively) before flattening**
>
> **Variant-2: always concatenate along the W-axis before flattening**
>
> **Variant-3: interleave tokens from right and left breasts (i.e., R1, L1, R2, L2, ... ; Rx/Lx means the x-th token in the flattened right/left sequence)**
>
> | Method             | Internal | Dice   | H.r.   | F.s.   | Sens.  | AUC    | External | Dice   | H.r.   | F.s.   | Sens.  | AUC    |
> | ------------------ | :------: | ------ | ------ | ------ | ------ | ------ | :------: | ------ | ------ | ------ | ------ | ------ |
> | **DR-M3D (ours)**  |    \|    | 0.6564 | 0.9228 | 0.9027 | 0.8073 | 0.9690 |    \|    | 0.5909 | 0.8974 | 0.8632 | 0.7207 | 0.9082 |
> | DRT-M3D Variant-1  |    \|    | 0.6536 | 0.9127 | 0.9083 | 0.8440 | 0.9693 |    \|    | 0.5808 | 0.9060 | 0.8729 | 0.8190 | 0.9206 |
> | DRT-M3D Variant-2  |    \|    | 0.6551 | 0.9039 | 0.9062 | 0.8532 | 0.9732 |    \|    | 0.5923 | 0.9060 | 0.8708 | 0.8476 | 0.9258 |
> | DRT-M3D Variant-3  |    \|    | 0.6414 | 0.9083 | 0.9018 | 0.8211 | 0.9650 |    \|    | 0.5749 | 0.8974 | 0.8644 | 0.7810 | 0.9162 |
> | **DRT-M3D (ours)** |    \|    | 0.6590 | 0.9229 | 0.9145 | 0.8578 | 0.9768 |    \|    | 0.5948 | 0.9117 | 0.8766 | 0.8762 | 0.9371 |
>
> Our strategy not only processes all tokens from one side before those from the other in the Mamba layer, but also maintains a roughly consistent distance between anatomically symmetric locations regardless of scanning order. This facilitates effective learning and comparison of bilateral features, as demonstrated in the results above.
>
> **W3**: concerns about the idea of feeding multi-view inputs to an attention-style model
>
> **A**: We appreciate the reviewer’s thoughtful comparison. We acknowledge that this work may appear analogous to multi-view approaches commonly seen in 2D mammography literature from the following two points:
>
> 1. Our framework explicitly leverages information from both left and right breasts, and takes them as two parts of the input.
> 2. The prioritized scanning along multiple axes in Mamba-3D is analogous to observing the input 3D data from multiple views.
>
> However, our method is fundamentally distinct from traditional multi-view settings. In our framework, all inputs (high/low-resolution patches of the left/right breast, as shown in Figure 3 of the main paper) are extracted from a single 3D NCCT image. Thus, our approach does not fuse data from independent acquisitions or views, but rather, analyzes different aspects (dual-resolution, left/right breasts) of the same 3D NCCT scan. The notion of “multi-view” here is therefore not directly comparable to classic multi-projection imaging as in mammography.
>
> We hope this clarifies the novelty of our approach and its difference from existing multi-view transformer methods.
>
> **W4**: the fairness of the evaluation of main results regarding pre-training
>
> **A**: Thank you for raising this important concern. However, there is a misunderstanding here.
>
> 1. Fairness in pre-training application:
>
>    * For baselines not compatible with MAE pre-training (mainly CNN-based models), we have already implemented a supervised pre-training regime: first, training with the segmentation task alone; followed by joint fine-tuning on segmentation and classification. Empirically, this approach greatly improved the baselines’ performance and outperforms the original implementation.
>    * For architectures capable of MAE pre-training (models with standard Transformer or Mamba layers), we conducted MAE pre-training without any extra data, followed by fine-tuning using exactly the same protocol as our proposed method. The only difference among these methods lies in model design.
>
> 3. Ablation evidence:
>    To further clarify, we present results for nnUNet without the supervised segmentation pre-training stage in the table below. This comparison demonstrates the benefits of this additional pre-training for methods that do not support MAE pre-training.
>
>    | Method             | Internal |  Dice  |  H.r.  |  F.s.  | Sens.  |  AUC   | External |  Dice  |  H.r.  |  F.s.  | Sens.  |  AUC   |
>    | ------------------ | :------: | ---- | ---- | ---- | ---- | ---- | :------: | ---- | ---- | ---- | ---- | ---- |
>    | nnUNet w/o Seg-Pre |    \|    | 0.5872 | 0.8952 | 0.7437 | 0.6606 | 0.9070 |    \|    | 0.5243 | 0.8718 | 0.7003 | 0.5524 | 0.8533 |
>    | nnUNet w/ Seg-Pre (as reported in the paper) |    \|    | 0.6259 | 0.9127 | 0.8548 | 0.7844 | 0.9536 |    \|    | 0.5668 | 0.8889 | 0.8173 | 0.6286 | 0.8889 |
>
> **Q1**: concerns about the training speed of the proposed method comparing with other baselines
>
> **A**: We provide a detailed comparison of GPU memory consumption, training time, and inference time for our method and all baselines in the table below, with “Training Time” representing the sum of pre-training and fine-tuning.
>
> | Method               | Training Memory (GB) | Training Time (h) | Inference Time (s) |
> | -------------------- | ------------------ | --------------- | ---------------- |
> | nnUNet               |        3.826         |       31.0        |       0.126        |
> | VNet                 |        3.489         |       28.9        |       0.112        |
> | swinUNETR            |        11.975        |       36.8        |       0.139        |
> | nnFormer             |        2.063         |       10.9        |       0.029        |
> | 3D UX-Net            |        8.754         |       67.0        |       0.251        |
> | MedNeXt              |        12.095        |       43.1        |       0.194        |
> | U-Mamba-Bot          |        6.105         |       44.4        |       0.196        |
> | U-Mamba-Enc          |        11.810        |       77.9        |       0.291        |
> | EM-Net               |        6.218         |       34.9        |       0.116        |
> | SegMamba             |        9.580         |       66.2        |       0.318        |
> | ViT-S/8              |        4.611         |       22.1        |       0.034        |
> | T-ViT-S/8            |        4.629         |       26.9        |       0.047        |
> | **DR-M3D/8 (ours)**  |        2.565         |       14.6        |       0.049        |
> | **DRT-M3D/8 (ours)** |        2.581         |       15.7        |       0.059        |
> | **DR-M3D (ours)**    |        9.187         |       30.4        |       0.143        |
> | **DRT-M3D (ours)**   |        9.202         |       34.2        |       0.171        |
>
> Notably, our approach does not incur a significant training or inference time burden relative to conventional architectures, making it both effective and practical for large-scale clinical deployment.

---

> ### Comment · Reviewer_y5XU · 2025-08-05
>
> I appreciate the effort and explanation in the rebuttal, which helps a lot on addressing my concerns to the paper.
>
> Overall, my major concerns has been solved. The new experiment results shows that the comparison made in the paper is generally fair and the computational speed is also relatively acceptable given no hardware-level optimization for Mamba architecture.
>
> I would still suggest to reframe the section about the multi-view feature concatenation in the final version, I see some other reviewer has also mentioned this issue. Current clarification and experiment can help address this issue.
>
> After reading other reviewer's comments, I still insists this paper is above the acceptance bar. The concerns about motivation of using Mamba is clearly explained and demonstrated via experiment in the paper. New results comparing with SoTA also shows a reasonable performance. Overall, I would suggest acceptance for this paper.

---

> > ### Author Response · Authors · 2025-08-05
> >
> > Thank you very much for your positive feedback and thoughtful comments on our rebuttal. We greatly appreciate your recognition of the merits and innovations in our work, as well as your constructive suggestions.
> >
> > Regarding the section on multi-view feature concatenation, we will revise and improve this part in the final version, with clearer clarification and supporting experimental results as provided in our rebuttal. Thank you again for your valuable feedback.

---

### Official Review · Reviewer_THvZ · 2025-07-02

**Clarity:** 3
**Significance:** 3
**Originality:** 2
**Rating:** 4
**Confidence:** 3

**Summary:**

The paper proposes Dual-Res Tandem Mamba-3D (DRT-M3D), a multitask framework for bilateral breast lesion detection and malignancy classification using non-contrast chest CT NCCT. The approach leverages a dual-resolution architecture to balance fine-grained segmentation details and global contextual features, combined with a tandem input mechanism that models bilateral breast regions jointly via Mamba-3D blocks. This design enables cross-breast feature interaction to exploit subtle asymmetries. Experiments on multi-institutional NCCT datasets demonstrate state-of-the-art performance in both tasks, outperforming UNet, SwinUNETR, and Mamba-based baselines. Key contributions include the unified multi-task framework, dual-resolution design with mutual fusion, tandem bilateral modeling, and validation on diverse clinical data.

**Questions:**

1.	How will the authors expand data collection to include cross-regional populations, and have preliminary results from diverse cohorts been tested?
2.	It is mentioned in the paper that the method relies on a pre-trained segmentation model (Pre-Stage) to locate the breast region. What is the impact on Main-Stage performance if this stage fails (for example, if the breast boundary is blurred in obese patients)?
3.	Are there specific metrics to validate detection of tiny abnormalities, which are critical for early diagnosis?

**Ethical Concerns:**

["NO or VERY MINOR ethics concerns only"]

**Final Justification:**

Thank you for your detailed response, which has resolved most of my concerns. I strongly recommend that the authors incorporate the discussion on different architectures from W3, as well as the Model Efficacy points raised by other reviewers, into the final version of the paper. After considering the comments from the other reviewers, I have decided to maintain my score.

**Limitations:**

yes

**Quality:**

3

**Strengths And Weaknesses:**

Strengths:
1、The dual-resolution design and tandem input mechanism for bilateral modeling are innovative, addressing the challenge of integrating fine-grained detection with global classification while leveraging breast asymmetry.
2、Experiments on three internal and one external dataset, including ablation studies on key components (dual-resolution, mutual fusion, tandem input), validate the approach’s effectiveness. The focus on opportunistic analysis of NCCT avoids additional radiation/cost, aligning with practical clinical needs.

Weaknesses:
1.	In the Pre-Stage of the paper (Sec. 3.1), the pre-trained segmentation model is used to crop the breast region. If the small lesions are lost in the pruning process, it may lead to the failure of subsequent detection. Non-contrast CT itself has low resolution, and the contrast of small lesions (such as microcalcifications) is insufficient, which may further increase the risk of missed detection.
2.	The article mentions the use of Delta visualization to compare DRT-M3D with DR-M3D models and highlights the advantages of the two-input mechanism. However, from the information provided, it seems that these visualizations mainly focus on the single-layer N=1 case, but it may also ignore the real behavior of the model under the multi-layer structure. Although FIG. 11 demonstrates the significant Delta response on the LR path, a comparative display for the HR path or other key features is inadequate. This can make it difficult for the reader to fully understand all the differences between the two models and their practical significance.
3.	The paper demonstrates that applying the tandem input concept to ViTs and CNNs (Appendix E, Table 10) is less effective. While this is a good result for the authors, the analysis could go deeper. The paper attributes ViT's failure to the excessive number of tokens from small patches. Is this purely a memory/computation issue, or a fundamental limitation in the attention mechanism's ability to handle such granularity? A more nuanced discussion on why Mamba's selective state-space mechanism is uniquely suited for this "tandem" sequence concatenation, beyond linear complexity, would elevate the paper's technical contribution.

---

> ### Author Rebuttal · Authors · 2025-07-29
>
> **W1 part1 & Q2**: concerns regarding the Pre-Stage (breast region cropping) of the pipeline
>
> **A**: Thank you for this important question. The Pre-Stage segmentation is designed solely to localize the breast region, which is a relatively simple task. In our experiments, the Pre-Stage achieved a high **Dice score of 0.9598**, demonstrating its accuracy and reliablity.
>
> To further minimize the risk of missing boundary lesions or subtle structures (e.g., in obese patients or cases with blurred borders), we adopted a conservative cropping strategy: the predicted bounding box is expanded by 2 voxels along the D-axis and 16 voxels along the H- and W-axes. This ensures that small lesions near the periphery are retained within the cropped region.
>
> Notably, the goal of the Pre-Stage is not to perform fine-grained segmentation but to exclude irrelevant anatomy (e.g., abdominal areas) and isolate the bilateral breast regions for downstream modeling. Across all training and test cases, we have not observed any missed-lesion failure due to Pre-Stage cropping. In future deployment, rare failure cases can be flagged for manual review or fallback processing.
>
> **W1 part2**: concerns about using NCCT for breast lesion localization
>
> **A**: We fully acknowledge that using NCCT for breast lesion detection is not a routine diagnostic process in clinical practice. Our purpose is to achieve breast lesion detection and classification on existing chest CT scans, such as those conducted for lung or heart screening. This approach introduces no additional radiation exposure or economic burden to patients, and can be referred to as opportunistic screening.
>
> With the clinical value ahead, we acknowleged that it is relatively difficult for doctors to do breast lesion detection and classification on NCCT due to the lackness of contrast between the lesion and normal tissues. By collecting a large number of data, we are commited to achieve a clinically practical performance on this task leveraging the power of well-designed models. Our experimental results demonstrate that the proposed approach achieves strong and promising performance on NCCT images.
>
> For medical imaging, we would also like to clarify two important points: First, microcalcifications are generally evident in x-ray based imaging modalities, including CT and digital radiography (DR)—even on non-contrast CT—but are typically not obvious in breast ultrasound images. Second, NCCT provides the same spatial resolution as contrast-enhanced CT; its main limitation lies in the limited contrast.
>
> **W2 part1**: concerns about using N=1 in Delta visualization
>
> **A**: Thank you for your insightful comment. We focused our Delta visualizations on the N=1 case primarily for clarity and interpretability: at a depth of N=1, the cross-breast (bilateral) interaction patterns introduced by our Tandem Input mechanism can be most directly and transparently visualized and understood.
>
> For higher N (e.g., N=6 as used in our main experiments), we have observed that cross-breast interactions not only persist but often become even more pronounced and widespread across different layers. However, in deeper models, these effects are distributed across multiple blocks, makes concise visualization in the paper challenging.
>
> As additional support, our quantitative results demonstrate that the benefits observed with the Tandem Input mechanism are effective across models with different depths: as shown in Table 9, even with N=1, our model outperforms the strong baseline, while increasing depth to N=6 yields further improvements, confirming the effectiveness of our design beyond the single-layer setting.
>
> We appreciate your suggestion and, if desired, are happy to provide additional Delta visualizations for multi-layer cases in the supplementary material to further illustrate these cross-breast effects in deeper models.
>
> **W2 part2**: clarification regarding Delta visualization on the HR path
>
> **A**: The Delta responses on the HR path mainly reflect the model’s focus on local textures and anatomical details, which aligns with its primary role for the segmentation task. Meanwhile, the LR path is designed to focus on capturing bilateral context with shorter sequence length when Tandem Input mechanism is used (i.e., in DRT-M3D), so the Delta visualization of the LR path shows clear cross-breast clues. Overall, the Delta visualization of the two paths in Figure 11 appropriately reflects the distinct roles of the HR and LR paths for their respective tasks.
>
> Compared to DR-M3D, which lacks the Tandem Input mechanism, DRT-M3D exhibits cross-breast responses in the LR path’s Delta visualization, further highlighting the effectiveness of our proposed design for modeling bilateral interactions, whereas in both models the HR path primarily focuses on local segmentation.
>
> **W3**: deeper analysis of why applying the Tandem Input concept to ViTs and CNNs is less effective than Mamba
>
> **A**: Thank you for this excellent question. We agree that the Tandem Input design’s impact varies fundamentally across architectures, and a deeper discussion is warranted.
>
> *Why does ViT struggle?*
>
> The limited effectiveness of Tandem Input on ViTs is not simply due to memory or computation. As token count increases, self-attention matrices become extremely large, leading to peaked and less expressive attention distributions. Although Transformers are theoretically capable of modeling arbitrary long-range dependencies, in practice—especially for very long sequences and without sufficient scaling of model and data size—the effective capture of such dependencies is often sparse and unreliable (Tay et al., 2021). Furthermore, ViT lacks inherent local inductive bias, making it harder to exploit local spatial structure compared to architectures with stronger locality, such as Mamba or CNNs.
>
> *Why does CNN struggle?*
>
> CNNs cannot directly model long-range bilateral dependencies, as their local receptive fields require progressively stacking many convolutional layers to cover larger spatial extents. Therefore, simply concatenating left and right views along the W-axis is not effective for explicit bilateral modeling. Although stacking these views as channels may seem like a reasonable alternative, breast symmetry is not strictly voxel-to-voxel; due to natural anatomical differences, positional variations, and imaging variability, this approach may mix features from non-corresponding areas and ultimately introduce negative effects.
>
> *Why is Mamba-3D effective?*
>
> Mamba-3D, in contrast, uniquely combines strong local bias and efficient selective long-range modeling, along with the support for extremely long sequences. With our Tandem Input mechanism, anatomically symmetrical parts of the left and right breasts always have a roughly fixed distance in the sequence (analogous to corresponding words in an antithetical couplet or rhyming lines), allowing the model to better capture cross-breast relationships.
>
> We appreciate the opportunity to clarify these architectural distinctions and will elaborate on these points further in the camera-ready version.
>
> **Q1**: preliminary results from diverse cohorts and expansion to cross-regional populations
>
> **A**: Thank you for raising this important point about the generalizability of our approach. The datasets used in our study were collected through collaborations with professional medical institutions, and we are actively expanding our partnerships to include hospitals from additional regions. Data collection and annotation from new cohorts are ongoing.
>
> To report preliminary results from more diverse cohorts, we have conducted a reader study using newly collected 100 cases from a separate region for the classification task. The table below summarizes the performance of our method (DRT-M3D) compared with two radiologists and a baseline method (nnUNet).
>
> ||Sens.|Spec.|AUC|
> |-|-|-|-|
> |Radiologist-1|0.75|0.76|-|
> |nnUNet|0.91|0.76|0.9334|
> |**DRT-M3D (ours)**|0.98|0.76|0.9864|
> |Radiologist-2|0.56|0.95|-|
> |nnUNet|0.76|0.95|0.9334|
> |**DRT-M3D (ours)**|0.93|0.95|0.9864|
>
> While it is important to note that radiologists typically do not assess for breast cancer on NCCT in routine clinical practice; nevertheless, our method achieves markedly higher performance. This demonstrates the robustness of our model and its potential value for opportunistic breast cancer screening.
>
> **Q3**: specific metrics to validate detection of tiny abnormalities, which are critical for early diagnosis
>
> **A**: Thank you for highlighting the importance of detecting tiny abnormalities for early diagnosis. To specifically address this concern, we evaluated our model on a subset (22%) of the test set consisting of cases with lesions less than 20 mm in diameter (corresponding to T1-stage breast cancer).
>
> The table below summarizes the performance of our methods and the baseline (nnUNet) on this subset.
>
> |Method (Spec.=0.9667)|Dice|H.r.|F.s.|Sens.|AUC |
> |-|-|-|-|-|-|
> |nnUNet|0.5393|0.9000|0.7995|0.5889|0.9173|
> |**DR-M3D (ours)**|0.5923|0.9222|0.8111|0.6556|0.9250|
> |**DRT-M3D (ours)**|0.6059|0.9333|0.8668|0.8333|0.9639|
>
> Although some metrics decrease slightly compared with the results on the full test set (which includes larger lesions; see Table 1 and Table 2 in the main paper), our model still achieves high Hit-rate, Sensitivity, and AUC on early-stage lesions. These results confirm the model’s practical value for the detection of tiny abnormalities that are critical for early breast cancer diagnosis.
>
> **References**
>
> [Tay et al., 2021] Tay, Y., Dehghani, M., Abnar, S., et al. Long Range Arena: A Benchmark for Efficient Transformers. International Conference on Learning Representations (ICLR), 2021.

---

> > ### Comment · Area_Chair_cCKV · 2025-08-02
> > **Rebuttal**
> >
> > Dear Reviewer THvZ
> >
> > Thanks for your review. Could you read the authors' rebuttal and update your review if needed?
> >
> > Best regards
> > Your AC

---

> ### Author Response · Authors · 2025-08-06
>
> Thank you for your time and effort in reviewing our paper. As there have been no further comments during the discussion period so far, we would like to check if there are any remaining questions or concerns we can address. Please feel free to let us know if any additional clarification is needed.

---

> ### Comment · Reviewer_THvZ · 2025-08-06
>
> Thank you for your detailed response, which has resolved most of my concerns. The detailed discussion on the architectural differences between Mamba, ViT, and CNNs is good. It provides a deeper understanding of why the Tandem Input mechanism is uniquely effective in their proposed framework. I strongly recommend that the authors incorporate the discussion on different architectures from W3, as well as the Model Efficacy points raised by other reviewers, into the final version of the paper.

---

> > ### Author Response · Authors · 2025-08-06
> >
> > Thank you very much for carefully reading our detailed rebuttal and for your positive feedback. We also appreciate your thoughtful consideration of the comments from other reviewers as well as our responses to them, some of which addressed misunderstandings raised during the initial review process. We are pleased that our responses have resolved most of your concerns.
> >
> > Regarding your suggestion to incorporate the discussion on different architectures from ***W3***, as well as the Model Efficacy points raised by other reviewers, we sincerely appreciate these valuable suggestions for further clarification and demonstration. We will integrate these improvements into the final version of our paper, presenting them in a clearer and more explicit manner.

---

> > ### Author Response · Authors · 2025-08-07
> > **Remaining Originality Concerns?**
> >
> > We appreciate your recognition of our approach's innovation in the Strength section. We have carefully noted that the Originality rating was marked as 2, which typically indicates a moderate level of novelty. As there do not appear to be any further concerns regarding originality in your comments, we hope you might consider whether a more favorable evaluation of originality would reflect your positive assessment.
> >
> > If there are any remaining concerns or suggestions, we would greatly appreciate further clarification or discussion.
> >
> > Thank you again for your time and consideration.

---

> > > ### Comment · Reviewer_THvZ · 2025-08-08
> > >
> > > Thank you for your continued feedback. I understand your concern regarding the Originality rating of 2. My overall evaluation of the paper is borderline accept, and the sub-ratings were carefully considered. I agree that the dual-resolution design and tandem input mechanism for bilateral modeling are innovative, and these contribute to tasks like bilateral breast lesion detection and malignancy classification. However, the judgment of the Originality rating should be based on a broader view of the Mamba architecture community and the field of medical imaging. I believe this method technically extends and improves existing techniques for specific tasks, rather than offering a fundamental breakthrough. Therefore, based on this perspective, I have maintained the Originality rating at level 2 (fair).

---

> ### Author Response · Authors · 2025-08-08
>
> Thank you very much for your prompt and detailed reply. We appreciate your clarification regarding the originality rating. At this point, we confirm that we have no further concerns on this matter.
>
> We fully agree with your perspective on the broader context of the Mamba architecture community and the field of medical imaging. We especially appreciate your aspiration for breakthrough work and dedication to advancing innovation. Our efforts are motivated by the desire to contribute meaningful progress within this dynamic landscape, and your recognition of our contribution is greatly valued.
>
> Thank you again for your valuable time and constructive feedback throughout the review process.

---

### Official Review · Reviewer_oeU9 · 2025-07-05

**Clarity:** 3
**Significance:** 3
**Originality:** 3
**Rating:** 4
**Confidence:** 4

**Summary:**

This paper presents a dual-resolution 3D Mamba framework for breast lesion detection and classification from early breast CT images. The authors propose a multi-task learning framework that combines pixel-level lesion segmentation with image-level diagnosis tasks, effectively integrating both local details and global features. Experiments conducted on internal and external datasets demonstrate the state-of-the-art performance of the proposed method.

**Questions:**

1. The authors should compare their method with more recent state-of-the-art (SOTA) approaches for tumor classification based on segmentation methods, including those mentioned in the weaknesses section.

For example, A Data-Efficient Pan-Tumor Foundation Model for Oncology CT Interpretation, https://arxiv.org/abs/2502.06171
FreeTumor: Large-Scale Generative Tumor Synthesis in Computed Tomography Images for Improving Tumor Recognition https://arxiv.org/abs/2502.18519

2. The novelty of the proposed method should be further emphasized. What are the unique aspects and motivations behind this approach compared to other Mamba-based methods and related techniques?

**Ethical Concerns:**

["NO or VERY MINOR ethics concerns only"]

**Final Justification:**

I really appreciate the effort and explanation in the rebuttal, which helps a lot on addressing my concerns regarding the novelty, especially the motivation of the Mamba architecture design and MTL training strategy. After reviewing the authors’ rebuttal and other reviewers’ comments, my concerns about the novelty and necessity of the work have been addressed. I will raise my score to weak accept. I recommend the authors include the following in the final version:

1) I recommend adding a discussion section to explore how the image modality relates to Mamba, including comparisons with different model backbones.
2) Since this is the first work to use non-contrast CT for breast lesion detection, I think it is necessary to include a figure comparing contrast and non-contrast CT. This will help highlight the added challenges of using non-contrast CT instead of contrast CT.

**Limitations:**

yes

**Quality:**

2

**Strengths And Weaknesses:**

Strengths:

1. This paper presents a novel Mamba-based framework to address the important problem of early breast lesion detection. The design of the tandem input, which incorporates both left and right breast images, is particularly innovative and interesting.
2. The authors conduct comprehensive experiments and present solid results across three datasets.
3. The paper is well-organized, and the quality of the English writing is commendable.

Weaknesses:

1. Missing comparsion with SOTAs: The authors did not compare with recent SOTA such as [1].

[1] Sun J, Xi X, Wang M, et al. A deep learning model based on chest CT to predict benign and malignant breast masses and axillary lymph node metastasis[J]. Biomolecules and Biomedicine, 2025.

2. Limited Novelty & Relations between task and architecture design: the author designed a framework specific for breast lesion detection and classification. The novelty of their framework seem only two fold: (1) using Mamba on breast lesion detection and classification (2) design a multi-task learning (classification and detection) strategy to obtain local & global features. However, regarding the first point, the author fails to thoroughly discuss the relation between mamba and this specific task, why should we use Mamba instead of other backbones? Is there any special characteristic of non-contrastive CT that mamba could excel at and outperform other backbones?
Regarding the second point, I think there are plenty of previous works that has design such multi-task learning strategy [2,3], just on different modalities or different lesions, they also design their multi-task learning framework for the same purpose.

[2] He Q, Yang Q, Su H, et al. Multi-task learning for segmentation and classification of breast tumors from ultrasound images[J]. Computers in Biology and Medicine, 2024, 173: 108319.
[3] Xu, Mengjuan et al. “A Regional-Attentive Multi-Task Learning Framework for Breast Ultrasound Image Segmentation and Classification.” IEEE Access 11 (2023): 5377-5392.

3. “Tandem input” is confusing: the definition of tandem input is a little confusing, is it just combine the feature sequence of left and right beast together as one unified sequence? I think the author should provide some references to give a better explanation of “tandem input”.

4. Missing comparison with contrast CT: I believe including comparison with using contrast CT as upper-bound could better reveal the clinical value of the proposed method.

5. Limited performance improvement: from Table.1, we can see that there is a very limited performance improvement comparing the proposed method and other very common baseline models, the author also didn’t compare methods that are specially designed for breast lesion classification.

6. Unfair comparison: the author conduct a large-scale self-supervised pre-training for their proposed method before finetuning, however, for their counterparts which are majorly from computer vision field, they did not employ the pretraining, which resulted in unfair comparison.

7. Unclear comparison in the visualization: In figure. 4, the authors present qualitative comparison results between their methods and other counterparts, however, the difference is too small and unclear. I think a zooming in and a highlight on the error part will do better.

8. Model Efficacy: since the topic is early detection, I believe detection speed and efficacy are both important, the author should add training time/inference time, and memory consumption.

---

> ### Author Rebuttal · Authors · 2025-07-29
>
> **W1 & Q1**: comparison with more recent SOTAs
>
> **A**:  We appreciate the reviewer’s suggestion on conducting more comparisons with recent SOTA. We have added the results of 5 recent methods mentioned by the reviewer, using the same metrics and datasets as Table 3 in the main paper.
>
> |Method|Internal|Dice|H.r.|F.s.|Sens.|AUC|External|Dice|H.r.|F.s.|Sens.|AUC|
> |-|:-:|-|-|-|-|-|:-:|-|-|-|-|-|
> |Sun et al. 2025|\||0.6153|0.9127|0.8548|0.7706|0.9444|\||0.5624|0.8974|0.8246|0.6286|0.8827|
> |He et al. 2024|\||0.5845|0.9039|0.7620|0.6789|0.9280|\||0.5176|0.8462|0.6271|0.6381|0.8540|
> |Xu et al. 2023|\||0.6069|0.9039|0.8526|0.7798|0.9566|\||0.5672|0.8974|0.8160|0.6952|0.9023|
> |Lei et al. 2025|\||0.6088|0.8996|0.8221|0.7890|0.9411|\||0.5658|0.8803|0.7967|0.7048|0.8931|
> |Wu et al. 2025|\||0.6140|0.9127|0.8384|0.7339|0.9512|\||0.5636|0.8547|0.8358|0.6952|0.9014|
> |nnUNet|\||0.6259|0.9127|0.8548|0.7844|0.9536|\||0.5668|0.8889|0.8173|0.6286|0.8889|
> |**DR-M3D (ours)**|\||0.6564|0.9228|0.9027|0.8073|0.9690|\||0.5909|0.8974|0.8632|0.7207|0.9082|
> |**DRT-M3D (ours)**|\||0.6590|0.9229|0.9145|0.8578|0.9768|\||0.5948|0.9117|0.8766|0.8762|0.9371|
>
> It should be noted that while some methods above are not designed for 3D data or CT modality, we have made every effort to adapt them to our 3D non-contrast CT setting (e.g., using Conv3D instead of the original Conv2D modules) for a fair comparison. The results show that our method has clear advantage over other methods. The strong performance of our method can be attributed to its Dual-Resolution and Tandem Input design, together with Mamba-3D's ability on local and global modeling, which is particularly beneficial for breast lesion detection and classification on NCCT. These merits are further explained in the answer to ***W2 & Q2***.
>
> **W2 & Q2**: concerns regarding task-architecture rationale and novelty
>
> **A**:  Thank you for the constructive feedback. We address these concerns from three aspects:
>
> 1. Why Mamba (Mamba-3D) is well suited to this task:
>
>    * NCCT images pose unique challenges—lesions often appear subtle, and bilateral asymmetries serve as key diagnostic cues. This calls for a model that can capture long-range context without sacrificing local detail.
>    * Mamba fits this need well. Unlike CNNs, which are less effective at capturing global context, and ViTs, which trade local detail for manageable sequence length with large patches, Mamba supports small patch decomposition and long-range modeling. This enables retention of fine spatial resolution (important for segmentation) while also capturing global or even cross-breast patterns (critical for classification).
>
>    * As for Mamba-3D, the cyclically prioritized scanning strategy further enhances local spatial modeling for 3D data. Unlike ViT, which often struggles to capture neighborhood relationships as sequence length increases, Mamba-3D enables more efficient learning and faster convergence on volumetric medical images.
> 2. Novelty of our multi-task design compared to existing methods:
>    * In previous studies, the classification branch was derived from the segmentation branch. This resulted in excessive feature sharing between these two tasks. Even though the tasks were related, it could still cause interference between features, leading to suboptimal performance.
>    * Our Dual-Resolution design explicitly balances dense (segmentation) and global (classification) tasks. By decoupling feature spaces while enabling interaction through Mutual Fusion, we reduce task interference and promote complementary learning, overcoming limitations seen in previous multi-task frameworks.
>    * The comparison results of the mentioned multi-task papers are provided in the table in ***W1***.
> 3. Additional novelty:
>    * Our “Tandem Input” strategy within the Mamba-3D framework enables effective cross-breast contextual reasoning. This is important in clinical diagnosis but has been rarely explored in 3D settings. This strategy has demonstrated improved classification AUC for NCCT-based breast cancer analysis.
>    * To our knowledge, this is the first framework to jointly perform breast cancer detection and classification on non-contrast chest CT scans, demonstrating the clinical potential of NCCT for breast cancer opportunistic screening.
>
> In summary, our work introduces targeted architectural and algorithmic innovations tailored to the unique challenges of breast cancer detection on non-contrast 3D CT. By combining effective sequence modeling for small 3D patches, decoupled multi-task design, and bilateral contextual reasoning, we achieve superior performance in both segmentation and classification.
>
> **W3**: a better explanation of “Tandem Input”
>
> **A**:  Thank you for pointing out the potential confusion around “Tandem Input”. “Tandem” is a term used to describe two or more things arranged one after another or working together in sequence. *For example, tandem bicycle refers to a bicycle with two seats and two sets of pedals, arranged one behind the other.*
>
> In the context of our work, “Tandem Input” refers to the concatenation of left and right breast token sequences into a unified input while *maintaining anatomical alignment*. This not only ensures that Mamba-3D scans all voxels from one breast before moving to the other, but also guarantees a roughly fixed distance between anatomically symmetric locations in the sequence (just like the corresponding words in an antithetical couplet or rhyming lines), allowing the model to better capture cross-breast relationships.
>
> Please refer to ***Reviewer y5XU’s W2*** for additional results of other mechanisms for constructing bilateral input sequences.
>
> **W4**: comparison with contrast CT
>
> **A**:  Thank you for highlighting the importance of comparison with contrast CT. To address this, we have supplemented our study with results based on contrast CT images which are used for initial tumor boundry delineation as mentioned in the Appendix B.
>
> The table below summarizes the performance of our method on corresponding contrast CT data, using the same metrics as Table 3 in the main paper.
>
> |Method|Internal|Dice|H.r.|F.s.|Sens.|AUC|External|Dice|H.r.|F.s.|Sens.|AUC|
> |-|:-:|-|-|-|-|-|:-:|-|-|-|-|-|
> |nnUNet|\||0.7078|0.9563|0.9323|0.8257|0.9724|\||0.6860|0.9487|0.9068|0.9238|0.9288|
> |**DR-M3D (ours)**|\||0.7252|0.9476|0.9436|0.9129|0.9736|\||0.6878|0.9345|0.9146|0.9302|0.9376|
> |**DRT-M3D (ours)**|\||0.7263|0.9534|0.9479|0.9480|0.9795|\||0.7018|0.9544|0.9337|0.9365|0.9497|
>
> As expected, contrast CT achieves higher performance, serving as a practical upper-bound for this task. Notably, our method on NCCT reaches performance (the last 3 lines of the Table in W1) that is close to that of contrast CT, demonstrating its effectiveness and potential clinical value—especially for large-scale screening workflows where contrast CT may not be feasible.
>
> **W5**: limited performance improvement and comparison with specially designed breast lesion classification methods
>
> **A**:
>
> 1. On performance improvements:
>
> We respectfully disagree that the improvement is limited. As shown in Table 1 and Table 2 of the main paper, our method achieves notable gains over competing methods on key metrics—even though all baselines were thoroughly optimized under identical protocols (same training set, data augmentation, preprocessing, and postprocessing). The table below provides a summary for Dice and AUC. It is notable that even modest improvements are valuable in clinically challenging tasks where metrics such as AUC are already near the ceiling.
>
> |Datasets\||Competing Methods' Dice\||Ours Dice\||Competing Methods' AUC\||Ours AUC|
> |-|-|-|-|-|
> |Inst. 1|0.5601 ~ 0.6231|0.6608|0.9361 ~ 0.9622|0.9909|
> |Inst. 2|0.5671 ~ 0.6548|0.6750|0.9382 ~ 0.9596|0.9743|
> |Inst. 3|0.4901 ~ 0.5980|0.6056|0.8792 ~ 0.9391|0.9471|
> |Inst. 4|0.5059 ~ 0.5776|0.5948|0.8777 ~ 0.9156|0.9371|
>
> 2. On comparison with specially designed breast lesion classification methods:
>
> We have included results for all representative methods mentioned by the reviewer, as detailed in our response to ***W1***. Notably, most “specially designed” breast lesion classification methods are variants of nnUNet or UNet, and are often developed on smaller datasets or with different imaging modalities from ours. In our study, we implemented and thoroughly tuned these methods for a fair comparison. As the results demonstrate, our approach outperforms all published competing methods, including those specifically tailored for breast lesion classification.
>
> **W6**: concerns regarding unfair comparison
>
> **A**:  Thank you for raising this important concern. We would like to clarify that all competing models were effectively pre-trained under comparable conditions: ViT and Mamba-based models underwent MAE pre-training, while CNN-based methods were pre-trained on dedicated breast and lesion segmentation tasks.
>
> Additionally, our pre-training does not qualify as “large-scale”, since we did not use any extra data.
>
> A similar concern was raised by ***Reviewer y5XU’s W4***; please see our detailed response there for further clarification.
>
> **W7**: unclear comparison in the visualization
>
> **A**:  We appreciate the reviewer’s suggestion regarding visualization clarity. We acknowledge that the differences may be difficult to appreciate in Figure 4. For the camera-ready version, we will address this by providing zoomed-in views and clear highlights (e.g., arrows) to draw attention to regions with notable differences.
>
> **W8**: concerns regarding model efficacy
>
> **A**:  Detailed comparisons of training time, inference time, and memory consumption are provided in our response to ***Reviewer y5XU’s Q1***, and are omitted here due to space constraints.
>
> In practical scenarios, all evaluated methods—including ours—can deliver predictions for each scan within tens to hundreds of milliseconds, fully meeting real-world throughput requirements. Our model already demonstrates very high computational efficiency.

---

> > ### Comment · Area_Chair_cCKV · 2025-08-02
> > **Rebuttal**
> >
> > Dear Reviewer oeU9
> >
> > Thanks for your review. Could you read the authors' rebuttal and update your review if needed?
> >
> > Best regards
> > Your AC

---

> > > ### Comment · Area_Chair_cCKV · 2025-08-06
> > > **Last chance for the discussion**
> > >
> > > Dear Reviewer oeU9
> > >
> > > As we are approaching the end of the discussion phase, it'd be great if you could have a look at the authors' rebuttal, interact with the authors and then reach a final decision.
> > >
> > > Best regards,
> > > Your AC.

---

> ### Author Response · Authors · 2025-08-06
>
> Thank you for your time and effort in reviewing our paper. As there have been no further comments during the discussion period so far, we would like to check if there are any remaining questions or concerns we can address. Please feel free to let us know if any additional clarification is needed.

---

### Author Response · Authors · 2025-08-09
**Summary of Rebuttal and Discussion**

We thank all reviewers for their thoughtful feedback and constructive suggestions, as well as the chairs for their organization and support throughout the review process.

As the discussion period draws to a close, we note that most reviewers have responded and acknowledged that our rebuttal has adequately addressed their concerns. In particular, both *Reviewer THvZ* and *Reviewer y5XU* indicated that they carefully considered the comments from other reviewers as well as our responses, and they expressed satisfaction with the clarifications and improvements. In light of these outcomes, we provide below a summary of the main issues resolved during the rebuttal and the resulting improvements made to our submission (not visible currently).

- **More Comparisons:**
  - Added results for all suggested SOTA methods (*Reviewer oeU9*).
  - Provided results on contrast-enhanced CT, confirming our method's clinical value and potential (*Reviewer oeU9*).
  - Added comprehensive efficiency analyses (runtime, training time, memory) (*Reviewer oeU9 & y5XU*).
- **Clinical Value and Evaluation:**
  - Emphasized the value of NCCT for opportunistic breast cancer screening, and demonstrated our method’s effectiveness in this setting (*Reviewer THvZ & 9b6M*).
  - Added a reader study on a new regional cohort to present preliminary cross-regional results (*Reviewer THvZ*).
  - Included results highlighting performance in early diagnosis (*Reviewer THvZ*).
- **Design Clarifications and Motivation:**
  - Clarified differences with previous multi-task (*Reviewer oeU9*) and multi-view (*Reviewer y5XU*) works.
  - Provided additional explanation and experiments for the Tandem Input mechanism and its advantages (*Reviewer oeU9 & y5XU*); explained why Mamba-3D is preferable to ViTs and CNNs (*Reviewer THvZ*); and offered further interpretation and results for the Delta visualization of Dual-Resolution paths (*Reviewer THvZ*).
  - Provided clear, intuitive explanations for all key methodological choices and innovations (Dual-Resolution, Tandem Input, Mamba-3D) (*Reviewer 9b6M*).
  - Explained how the reliability of the Pre-Stage was ensured (*Reviewer THvZ*).
- **Corrections and Response to Misconceptions:**
  - Clarified pre-training protocols across all competing methods (*Reviewer oeU9 & y5XU*).
  - Improved figures with highlights (*Reviewer oeU9*) and added high-level schematics (*Reviewer 9b6M*).
  - Made several minor corrections, including the previously missing reference for Figure 1 (*Reviewer 9b6M*).

We thank the reviewers once again for their constructive input, which has greatly improved the quality and presentation of our work. Based on the feedback received—including acknowledgement from those reviewers who responded that they have considered all concerns raised by all reviewers—we are pleased that all major issues have been addressed to the extent possible.

---

### Note · Authors · 2025-08-12

We would like to once again thank the reviewers and chairs for their dedicated efforts during the review process of our submission. As we have already provided a summary of the rebuttal and discussion phases, we take this final opportunity to succinctly highlight the core strengths and contributions of our work—especially those underscored by the reviewers.

**1. Novel Task and Practical Significance**

To our knowledge, this is the first study to demonstrate feasible, high-performance *breast cancer detection and classification using non-contrast chest CT (NCCT) scans*—unlocking a new avenue for opportunistic screening without extra cost (THvZ, y5XU, 9b6M). Although NCCT poses considerable challenges for this task, these very obstacles highlight its significance and practical value, offering the potential to extend screening access to underserved populations.

**2. Thoughtfully Designed Methodology**

Our framework is grounded in intuitive, principled design choices that address challenges on NCCT-based breast cancer analysis, rather than repurposing generic architectures—a highly innovative approach (oeU9, THvZ, 9b6M). In particular:

- The *Dual-Resolution* design explicitly balances dense and global tasks, addressing limitations of previous multi-task frameworks (oeU9, THvZ);
- The *Tandem Input* mechanism enables explicit cross-breast reasoning, broadening the possibilities for this challenging NCCT scenario (oeU9, THvZ, y5XU);
- Our customized use of *Mamba-3D* achieves both efficient local/global modeling and effective integration of bilateral breast inputs, superior to ViTs and CNNs (THvZ, y5XU).

**3. Clinical Value and Robustness**

Comprehensive experiments validate both the feasibility of this novel task and the superiority of our proposed method (oeU9, THvZ, y5XU, 9b6M). Our method demonstrates robust performance across multi-institutional datasets, consistently surpassing all known and suggested SOTA methods. Its clinical advantage and value are further substantiated by outstanding early-stage detection results, validation through reader studies on a cross-regional cohort, and performance that closely approaches that of contrast-enhanced CT.

In summary, our work introduces a practical and well-founded solution to an important but underexplored clinical application, supported by extensive validation and insightful reviewer feedback. We again thank the committee for their dedication and careful consideration.

---

### Decision · Program_Chairs · 2025-09-17

**Decision:**

Accept (poster)

**Comment:**

(a) Scientific Claims and Findings
The paper proposes Dual-Res Tandem Mamba-3D (DRT-M3D), a novel multitask framework for opportunistic breast lesion detection and malignancy classification using non-contrast chest CT (NCCT) scans. The model integrates:
- A dual-resolution design, balancing fine-grained segmentation with global contextual classification.
- A tandem input mechanism for joint bilateral modeling, capturing subtle asymmetries between left and right breasts.
- A multi-task framework, DRT-M3D, for lesion detection and breast-level classification for opportunistic breast cancer analysis on non-contrast
chest CT scans.
Experiments across four multi-institutional datasets demonstrate state-of-the-art results, surpassing UNet, SwinUNETR, ViTs, and other baselines. Ablations confirm the importance of the dual-resolution, tandem input, and Mamba-3D components. The paper also shows performance approaching that of contrast-enhanced CT, supporting its potential for opportunistic breast cancer screening without extra radiation or cost.

(b) Strengths
- First demonstration of high-performance breast cancer detection and classification using NCCT, enabling opportunistic screening.
- Dual-resolution and tandem bilateral modeling are well-motivated, addressing challenges of NCCT.
- Strong results across multi-institutional datasets, including cross-regional reader studies and early-stage cancer detection.
- Extensive ablations, comparisons to recent SOTAs, efficiency analyses, and visualizations.
- Paper is well-written.

(c) Weaknesses
- Some reviewers noted that multi-task learning and bilateral modeling have precedents in other modalities, so the contribution can be considered to be more on the incremental side than a fundamental breakthrough.
- Figures (e.g., Delta visualization, qualitative comparisons) were initially unclear or insufficiently highlighted.
- Questions were raised about whether improvements derive from larger/more tailored pre-training versus architectural design. Authors later clarified fairness across baselines.
- Although FLOPs are lower, practical runtimes were initially slower than optimized Transformers. Authors clarified this and showed near-real-time performance.
- Concerns about potential failure in localizing small lesions in NCCT (though authors showed safeguards with expanded cropping).
- While some found the tandem input and dual-resolution design innovative, others (THvZ) judged the originality only “fair” in light of broader Mamba/medical imaging trends.

(d) Decision Rationale
The paper makes a meaningful, clinically valuable contribution by introducing the first effective framework for breast cancer detection and classification using NCCT. While some architectural components adapt existing ideas, the integration is thoughtful, technically solid, and experimentally validated. The clinical impact and robustness of the evaluation outweigh concerns about limited novelty. I recommend the paper to be accepted as poster because, although impactful, the method is primarily an adaptation/extension of existing components rather than a breakthrough in core machine learning methodology. Its main strength lies in application significance and careful engineering.

(e) Rebuttal and Discussion
- SOTA comparisons: Authors added results for five recent methods and adapted them for 3D NCCT. Their model outperformed all, addressing this concern.
- Contrast CT as upper bound: Additional experiments confirmed DRT-M3D on NCCT approaches contrast CT performance.
- Efficiency concerns: Authors provided training/inference time and memory analysis, showing near-real-time applicability.
- Novelty and rationale: Authors clarified why Mamba-3D is well-suited for NCCT (local + global modeling, bilateral asymmetry handling).
- Reviewers acknowledged improved clarity; oeU9 upgraded to Weak Accept.
- Tandem input explanation: Authors gave detailed clarifications and additional experiments with variants. Reviewers agreed this addressed initial confusion.
- Pre-Stage segmentation reliability: Authors showed high Dice and conservative cropping strategies, reducing missed-lesion risk.
- Most concerns were satisfactorily resolved in rebuttal. The only remaining hesitation relates to perceived incremental novelty (THvZ), but reviewers overall leaned positive, with y5XU recommending acceptance and oeU9 upgrading their score.